# A Narrative Review of Cytokine Networks: Pathophysiological and Therapeutic Implications for Inflammatory Bowel Disease Pathogenesis

**DOI:** 10.3390/biomedicines11123229

**Published:** 2023-12-06

**Authors:** Marek Vebr, Renáta Pomahačová, Josef Sýkora, Jan Schwarz

**Affiliations:** Departments of Pediatrics, Faculty Hospital, Faculty of Medicine in Pilsen, Charles University of Prague, 323 00 Pilsen, Czech Republic; pomahacovar@fnplzen.cz (R.P.); sykorajo@fnplzen.cz (J.S.); schwarzj@fnplzen.cz (J.S.)

**Keywords:** inflammatory bowel disease, cytokines, novel therapeutic targets

## Abstract

Inflammatory bowel disease (IBD) is a lifelong inflammatory immune mediated disorder, encompassing Crohn’s disease (CD) and ulcerative colitis (UC); however, the cause and specific pathogenesis of IBD is yet incompletely understood. Multiple cytokines produced by different immune cell types results in complex functional networks that constitute a highly regulated messaging network of signaling pathways. Applying biological mechanisms underlying IBD at the single omic level, technologies and genetic engineering enable the quantification of the pattern of released cytokines and new insights into the cytokine landscape of IBD. We focus on the existing literature dealing with the biology of pro- or anti-inflammatory cytokines and interactions that facilitate cell-based modulation of the immune system for IBD inflammation. We summarize the main roles of substantial cytokines in IBD related to homeostatic tissue functions and the remodeling of cytokine networks in IBD, which may be specifically valuable for successful cytokine-targeted therapies via marketed products. Cytokines and their receptors are validated targets for multiple therapeutic areas, we review the current strategies for therapeutic intervention and developing cytokine-targeted therapies. New biologics have shown efficacy in the last few decades for the management of IBD; unfortunately, many patients are nonresponsive or develop therapy resistance over time, creating a need for novel therapeutics. Thus, the treatment options for IBD beyond the immune-modifying anti-TNF agents or combination therapies are expanding rapidly. Further studies are needed to fully understand the immune response, networks of cytokines, and the direct pathogenetic relevance regarding individually tailored, safe and efficient targeted-biotherapeutics.

## 1. Introduction

Inflammatory bowel disease (IBD) encompasses Crohn’s disease (CD) and ulcerative colitis (UC). IBD is a chronic relapsing immune-mediated disease that is likely to occur in early childhood to beyond the sixth decade of life, and is unfortunately incurable. Previous systematic reviews described rising incidence and prevalence of IBD among both children and adults around the world, and data are emerging from regions where it was previously thought to be uncommon [1,2]. The origin of this disease is not entirely clear, and several involved mechanisms have been postulated, such as genetics; defects in a number of cellular pathways, including the dysregulation of homeostasis, loss of epithelial barrier integrity, and tolerance to the gut microbioma; and environmental exposures, among other processes [3,4,5,6,7,8]. The immunological dysregulation in IBD is characterized by defects in the barrier functions and a failure of immune regulation to control the inflammatory response, and as a consequence of the breakdown of these pathways contributing the host–microbe dialogue, a chronic inflammatory response in the gut is triggered, leading to the release of pro- and anti-inflammatory cytokines [9].

Recent technological advances have led us into an ‘omics’ era in which it is reasonably cost-effective and almost routine to obtain genomic, transcriptomic-, proteomic-, and metabolomic-scale data, even from single cells. Multiple findings indicate that the recessive inheritance of rare and single-cell analyses of tissues affected by CD unveil heterogeneity among intestinal intraepithelial T cells and shifts in subset distributions [10]. Multiomics technologies enable the quantification of thousands of molecules and can provide new insights into the molecular landscape of immune-mediated diseases [11]. During the era of a number of biological therapies, applying multiomics analyses represents a promising frontier for exploring the intricate network of IBD pathogenesis, especially in the age of omics approaches and cutting-edge technologies [12]. The integration of multiomics data and deep phenotyping may enable the prediction of cytokine responses and detecting this pathway response [13]. Genome-wide associated studies (GWASs) identified genetic variants of a trait that convey IBD susceptibility with high confidence and their downstream signaling that have been used to obtain the specific molecular events that regulate the production of cytokines [14]. For example, loss-of-function mutations in the genes encoding interleukin-10 (IL-10) and the IL-10 receptor (IL-10R) are associated with early-onset IBD [15]. The nucleotide-binding and oligomerization domain-containing 2 (NOD2) genomic biomarker has been well studied in this regard. Notably, variation in NOD2 and additional risk factors could together be responsible for CD development [16]. Multiple findings indicate that the recessive inheritance of rare and low-frequency deleterious NOD2 variants contributes to 7–10% of CD cases, establishing NOD2 as a Mendelian disease gene associated with early-onset CD [17]. In future research, it is advisable to embrace a multiomic data approach, incorporating diverse sets of information encompassing clinical parameters, environmental exposures, genetics, epigenetics, immune function, and microbial structure [8].

Several pathways are proposed to drive disease [18]. The overall effect of an inflammatory response in IBD is dictated by the balance between the key pro- and anti-inflammatory cytokines [7,19]. This cytokine–cell network has been shown to play essential roles in cell signaling and the initiation and perpetuating of intestinal inflammation [20]. Depending on the IBD type and context, there continues to be a rapid expansion in the knowledge of how particular cytokine networks drive distinct features and phase of IBD and provide a basis for potential alternative therapeutic targets of these enigmatic entities [21,22]. Fortunately, the advent of the molecular biology revolution with the cloning of cDNAs for cytokines provided the necessary research tools for this research in IBD. This led to a new concept of the TNF-dependent cytokine cascade, cytokine dysregulation and targeted therapy [23,24]. Cytokine responses have been considered critical in driving intestinal inflammation in IBD in humans, and have become one of the successful targets of pharmaceuticals. The armamentarium has expanded to biologicals that can effectively target cytokines, such as tumor necrosis factor alpha (TNF), interleukin (IL)-12p40/IL-23 (targeting the common subunit of IL-12 and IL-23), or inflammatory cell recruitment with α4β7 blockers [25]. However, despite these successes, IBD still poses major therapeutic challenges, especially for therapy-refractory subjects requiring alternative therapeutic approaches. Innovations in bioengineering have aided in advancing our knowledge of cytokine biology and yielded new technologies for cytokine engineering [26].

Here, we extensively analyzed the effect in vitro, and in a preclinical therapy study in vivo, the correlations and the role of selected cytokines that are substantial for inflammatory reactions of IBD, nevertheless understanding the factors that regulate cytokine networks and signaling enabled cytokine-targeting therapies in the clinic. Thus, the remainder of the review focuses on recent human studies summarizing the latest trends of possible therapeutic targeting of approved and investigational cytokine-based therapy to the latest cutting-edge technology towards novel potential therapeutic targets.

## 2. Pathological Involvement of Multiple Cytokine Networks and T-Cell Subsets in IBD

Cytokines, which comprise of a family of small proteins (usually smaller than 30 kDa)—interleukins, interferons, chemokines, and numerous other mediators—are important components of the immune system. In Table 1, we can see cytokine groups and subgroups, and their effect in IBD. Our knowledge of immune-mediated inflammation has been constantly increasing over the last few decades due to a deeper understanding of cytokine networks that lie behind their pathophysiology. Cytokines are made mainly by helper T cells (Th) and macrophages that have been considered critical in the initiation, maintenance, and resolution of immune responses and cell-to-cell signaling [27]. Various pathological disorders stem from an imbalance in cytokine production, cytokine receptor expression, and/or the dysregulation of cytokine processes [28]. The signaling pathways and cytokine–cytokine receptor interaction pathways play important roles in the pathogenesis of CD [29]. Cytokines and their clinical significance are introduced from the perspective of their pro- and anti-inflammatory effects in the progression of IBD. A prominent factor is the cytokine’s pleiotropy nature, by which a given cytokine can induce differential, even opposite cell responses [30]. Therefore, cytokines, categorized into distinct groups based on their structural biology and associated signaling pathways, constitute one of the most crucial classes of biomolecules for comprehending real-time IBD biology [31]. Cytokine networks act in concert with specific cytokine inhibitors and soluble receptors on target cells to regulate the immune response, gut inflammation, and paracellular permeability; upregulate epithelial proliferation; and trigger restitutive processes. Pattern recognition receptor (PRR) signaling is carefully regulated, especially with respect to downstream cytokine secretion [32]. Cytokine interactions with receptors trigger signaling pathway activity to form a network fundamental to diverse immune processes, including host homeostasis and increasing inflammation in IBD [20]. Although the etiology of IBD has not been fully elucidated, data gathered from human studies on analysis of tissue samples from large IBD patient cohorts and mouse models of colitis summarized that the gut mucosa is both a source as well as a target for numerous cytokines, and that such signaling can substantially influence the outcome of mucosal disease, such as IBD [19,33]. Nevertheless, a direct comparison between in vitro and in vivo data, based on the same cytokine line, also reveals that major contributors to inflammation in vitro may not necessarily be of similar relevance in vivo situation, and vice versa.

The gut barrier that constitutes an important defensive line for the human body against the environment is patrolled by dynamic interactions between intestinal mucosa and innate and adaptive immune systems. It has been unanimously consolidated that the breakdown of the intestinal epithelial barrier is a characteristic feature of IBD [35,36], and cytokine-mediated immune–epithelial crosstalk that integrates the crosstalk of epithelial cells with innate and adaptive immune cells maintains intestinal homeostasis. However, the IL23/IL17 pathway has important roles in epithelial cell regulation. The Toll-like receptor (TLR) signaling and activation of NF-κB results in increased transcription of proinflammatory cytokines such as TNFα, LIGHT, IL-1β, and IL-6 and disruption of the intestinal barrier function [37]. The breakdown of the intestinal lining function, altered immune cell reactivity to intestinal microbiota, or inappropriate or exaggerated T-cell responses can lead to chronic inflammation and the tissue destruction characteristic of CD and UC. For these reasons, these properties suggest that IL-12 family cytokines have a key role in the regulation of intestinal homeostasis, and ultimately, the pathogenesis of IBD, and they have become potential targets for inhibiting the pathogenesis of inflammatory bowel disorders [38,39]. Cytokines that are constitutively active and consistently shuttle among various compartments of the intestinal mucosa have the capability to influence the division of epithelial cells and orchestrate the assignment of appropriate immune cells, establishing feedback loops. The immune cells in the gastrointestinal tract are predominantly localized in gut-associated lymphoid tissues such as Peyer’s patches, lymphoid follicles, and cryptopatches [40]. The intestinal mucosa consists of a meticulously organized epithelium that serves as a robust physical barrier against detrimental luminal contents, all the while facilitating the absorption of essential nutrients and solutes [41]. Intraepithelial cells (IECs) integrate both positive and negative interactions from the gut-residing microbiota, signaling neighboring immune cells to adapt to the microbiota. This process perpetuates the normal function of the body. Similar to immune cells, IECs have the capacity to secrete cytokines, chemokines, and growth factors [40,42]. A vital role of IECs is to uphold the integrity of the intestinal barrier. This function permits the passage of essential ions, nutrients, and water while inhibiting the entry of bacterial toxins and pathogens [43]. The maintenance of renewal of IECs requires tight regulation to avoid any imbalance in homeostasis [44].

Increased intestinal permeability and dysfunctional barrier have been recognized as a major and early feature of progression of IBD [45]. In CD subjects in apparent remission, increased permeability during remission may predict disease reactivation [46]. Immune responses in the gut and permeability of the barrier are tightly regulated via balance between gut-resident cells that promote host defense and those that suppress inflammation. Disruption of this balance and dynamic remodeling of cytokine networks during progression of IBD can lead to chronic intestinal inflammation characteristic of IBD. Classic inflammatory responses are triggered by pattern recognition receptors such as TLRs and nucleotide-binding oligomerization-domain protein (NOD)-like receptors (NLRs). TLR/NLR signaling results in the secretion of potent proinflammatory mediators, such as cytokines and chemokines. For example, cytokine networks can act positively or negatively, e.g., monocellular phagocytes (MNPs) are able to signal IL-10 during colonic damage that helps in restoration of homeostasis, but also IL-1β or TNF-α CD; thus, increased permeability during remission may be a risk factor for disease reactivation. STAT3-inducing cytokines, in particular IL-22 and IL-6, together with IL-17, promote survival of IECs and antimicrobial defense. Furthermore, IL-10 and TGF play a key role in promoting tolerance through MNP-T reg cell interactions [24].

Recent studies have revealed that the IL-1β-induced elevation in intestinal permeability plays a significant role in enhancing intestinal inflammation. This effect is mediated through regulatory signaling pathways and involves the activation of the nuclear transcription factor nuclear factor-κB, the activation of the myosin light chain kinase gene, and post-transcriptional modulation of the occludin gene by microRNA. Collectively, the modulation of the barrier by IL-1β in the context of gut inflammation represents a potential therapeutic target for addressing defective intestinal barrier function [32].

There are many conflicting data in the literature on how specific cytokines like IL-33 guide pro- and anti-inflammatory responses [40]. Crawford and colleagues suggest that the inflammatory cytokines TNFα and IFNγ directly induce intestinal barrier dysfunction and alter the tight junctions and rate of cellular turnover in bovine intestinal epithelial cells [47]. TNFα and IFN-γ are widely recognized for their indisputable role in regulating tight junction integrity. In vivo, Caveolin-1-dependent occludin endocytosis is essential for the tight junction regulation induced by TNF [48]. Meyer and colleagues present evidence demonstrating the association of TNFα, IFNγ, and IL-1β with elevated intestinal epithelial permeability, observed both in vitro and in vivo. IL-10, the most extensively studied cytokine, can induce anti-inflammatory mechanisms and is strongly linked to a protective function against disruptions in the gut barrier, particularly in situations associated with epithelial hyperpermeability. On the other hand, more controversial data have been reported for IL-6, IL-17, IL-22, IL-23, and IL-33 in this context [41]. Discoveries of new mRNA stabilizers and receptor-directed mRNA metabolism have provided insights into the means by which IL-17 cooperates functionally with other stimuli in driving inflammation, whether beneficial or destructive [49]. Working in conjunction with the mucosal layer, the epithelial layer engages in two-way communication with underlying immune cells to finely regulate the inflammatory response against bacterial toxins. This collaboration, along with specialized cells, establishes a well-equipped, intricately regulated, and stringent barrier continuously monitored by immune cells to foster an immune-silent environment to preserve intestinal homeostasis. Increasing evidence furthermore suggests that IL-23-independent IL-17 production regulates intestinal permeability [50] and bacteroidales recruit IL-6-producing intraepithelial lymphocytes in the colon to promote barrier integrity [51]. Experimental models of colitis have underscored the critical interference of IL-9-producing T cells with an intact intestinal barrier function, influencing cellular proliferation and tight junction molecules. The inhibition of IL-9 has been identified as a significant factor in ameliorating disease activity and severity in animal models of IBD. This suggests that targeting IL-9 could serve as a novel and focused approach for therapy [52].

Disruption of the homeostatic balance of intestinal dendritic cells (DCs) and (MNPs) may contribute to IBD. While the development and pathophysiology of the gut is governed through T-cell-associated activation pathways and cytokines networks, as cytokines are key mediators of cellular interactions in IBD, the role of MNPs is less understood. Dysregulated IECs and MNPs may precipitate the chronic inflammation in IBD. Sustained activation of innate responses can drive pathogenic T-cell responses, including the recruitment of MNPs, which produce key proinflammatory cytokines (IL-1β, IL-18, and TNF), which promote pathologic T-cell responses at the expenses of the Treg cell [24,53]. MNPs play a critical role in integrating microbial clues to promote a Treg cell response; CD4+T cells appear to be a key source. Two distinct Treg populations, Foxp3- TR1 cells and Foxp3+ Treg cells, suppress colitogenic T-cell responses through the production of IL-10 [53]. STAT-3-inducing cytokines IL-22, IL-6, and IL-17 promote the survival of IECs and antimicrobial defense. Additionally, IL-10 and TGF play a key role in promoting tolerating programs in MNPs, which can in turn drive Foxp3+ Treg cells [24]. Subsets of MNPs exhibit enrichment in the inflamed colons of individuals with IBD, and this enrichment aligns with the severity of the disease. Notably, these specific subsets of MNPs are also identified among the top enriched cell types in samples from individuals who do not respond to treatment with either infliximab or vedolizumab [54]. Defining the landscape of MNPs provided evidence for the expansion of CD163+ Mono/MΦ-like cells in UC only, highlighting a distinction between UC and CD, and thus the potential contribution of monocyte-like cells in driving colitis [55]. Chapuy et al. established a connection between monocyte-like CD163- MNPs, IL-12, IL-1β, and the identification of colonic memory IL-8-producing CD4+ T cells. These factors collectively may contribute to the pathogenesis of UC. In patients with UC, IL-12 and mucosal CD14+ monocyte-like cells induce IL-8 in colonic memory CD4+ T cells, whereas this effect is not observed in patients with CD [56].

The most convincing evidence for a potential dysregulation and maintaining the balance of the immune response and cell interactions also stems from the observation of the mucosal cellular populations and a vast array of cytokines associated with IBD [57,58,59]. High-dimensional single-cell profiling approaches, such as single-cell RNA sequencing (scRNA-seq), have recently been employed in the analysis of intestinal specimens from patients with IBD. These analyses offer unbiased insights into cell lineages and their functional states, deconvolute pathways underlying IBD pathogenesis, and provide biomarkers that can predict the course of the disease and the response to therapy [60]. A major conceptual advancement in our understanding of the pathobiology of IBD has been the realization that the interplay between immune cells and nonimmune cells and signaling pathways are substantially involved in the dynamic regulation of chronic gut inflammation regarding real-time IBD biology. However, not all cytokines produced within the damaged gut appear to play a crucial role in amplifying and perpetuating the inflammatory cascade associated with inflammatory bowel disease (IBD). The conceptual framework of the mucosal cytokine network has evolved over the years, moving from a Th cells dichotomy (Th1/Th2) to the balance between effector and regulatory T cells. Nowadays, the significance of myeloid cell instruction of lymphocytes, particularly through IL-12 and IL-23 (p19/p40), is increasingly recognized. Groundbreaking changes in patient care have been ushered in by anti-IL-12p40 agents, such as ustekinumab, and anti-IL-23p19-directed approaches are on the verge of significant success [61]. Cytokines belonging to the IL-1 family, such as IL-1β and IL-18, play a vital role in maintaining homeostatic conditions in the intestine, as mentioned earlier. The release of these two cytokines is predominantly reliant on the activation of the inflammasome complex. IL-1β secretion is driven by specific stimuli, while IL-18 is constitutively expressed by the intestinal epithelium [44,62]. Yet, their precise role is not always clearly defined, and many questions remain concerning the role of specific cytokines in different types of IBD within distinct regions of the gut. Nevertheless, downstream of these multiple interactions and response is essential for producing the IL-1 family cytokines. If dysregulated, this immunomodulatory function of epithelial cells and defects in such pathways might contribute to the cytokine pathways initiating intestinal inflammation and initiation of IBD [63]. The existence of a complex network of soluble mediators and a simultaneous release of pro- and anti-inflammatory cytokines are mandatory in any immune response that has important implications for disease progression. The existence of a plethora of regulatory cytokines secreted by activated lamina propria that has important implications for inflammation progression has been reported thus the imbalance between proinflammatory and anti-inflammatory cytokines in IBD hinders the resolution of inflammation and, instead, contributes to the perpetuation of the disease and tissue destruction, in particular, the imbalance between proinflammatory (TNF, IFN-γ, IL-1β, IL-6, IL-12, IL-21, IL-23, IL-17, integrin, etc.) and anti-inflammatory cytokines (IL-10, TGFβ, IL-35, etc.) [64].

Gut-resident T cells—and in particular CD4+ Th cells, which reside primarily in the lamina propria in the basal state—play a significant role in the relapsing and remitting course and persisting low-grade inflammation, specifically in IBD [65]. T cells are generated following activation of CD4+ helper cells through the mechanisms underlying CD4+ T-cell differentiation, including cytokine-induced signaling and transcriptional networks [66]. The basic field of polarized specific immune responses mediated by CD4+ T helper (Th) lymphocytes is based on their profile of cytokine production (type 1 or Th1 and type 2 or Th2). Well-known T-cell subsets encompass T helper (Th)1, Th2, Th9, Th17, Th22, T follicular helper (Tfh), and various types of T-regulatory cells (Treg). T cells are generated in response to, and adapt to, microenvironmental conditions. They participate in a complex network of interactions with other immune cells, influencing the further progression of IBD [67,68,69]. Intraepithelial lymphocytes (IEL) encompass various unique T-cell subsets, including NKp30+γδT cells expressing RORγt and producing IL-26 upon NKp30 engagement. Further analyses, comparing tissues from noninflamed and inflamed regions of patients with Crohn’s disease (CD) versus healthy controls, reveal increased activated Th17 but decreased CD8+ T, γδT, TFH, and Treg cells in inflamed tissues. Similar analyses found increased CD8+, as well as reduced CD4+ T cells with an elevated Th17 over Treg/Tfh ratio. These examinations of CD tissues suggest a potential link, pending additional validations, between transmural inflammation, reduced IEL γδT cells, and altered spatial distribution of IEL and T-cell subsets [10]). Further information on these cells can be found in other reviews [67,70].

High levels of inflammatory cytokines, including TNF-α, IL-1β, IL-6, IL-17, IL-22, and IL-23, can drive IBD intestinal inflammation [71]. It is commonly believed that CD is usually driven by Th1/ Th17 dominated response with an upregulation of IL-12 family cytokines including IL-23, IFN-γ, and IL-17, whereas UC, in contrast, is mostly characterized by excessive Th2/Th9 response with increased levels of IL-13, IL-5, and IL-9, playing a critical role in disease mechanisms. IL-23 has been shown to be associated with both CD and UC pathology. The distinction between cytokine subsets is somewhat arbitrary, as all cytokines are produced in the inflamed mucosa, albeit in vastly different proportions. Tissue damage is likely mediated primarily by nonpolarized proinflammatory cytokines, such as IL-1β, IL-6, IL-8, and TNF [15]. IL-12 and IL-23 regulate the differentiation of Th1 and Th17 cells, and, along with IL-27 and IL-35, play a crucial role in the balance of inflammatory immune responses. Th17 cells are a subset of CD4+ T cells characterized by the secretion of IL-17 and expression of a nuclear transcription factor, retinoic acid receptor-related orphan receptor gamma t (RORγt) [72]. Moreover, optimal induction of Th17 cells occurred with the combination of TGFβ, IL-1β, IL-6 via mediating phosphorylation of STAT3, which is further amplified by signaling from IL-23 and IL-21 in a positive feedback loop [73,74,75]; similarly, IL-4 drives Th2 cell differentiation, which are characterized by expression of GATA3 and production of IL-4, IL-5, and IL-13 [76]. Disease-specific cytokine patterns give rise to a second tier of cytokines that bridge the Th1/Th17–Th2 divide, serving as both upstream facilitators and downstream mediators of inflammation. This group includes well-known cytokines such as TNF-α, IL-1β, and IL-6, along with a more recently studied cytokine called TL1A [77]. Numerous studies have reported elevated expression of Th17 pathway cytokines, including IL-1β, IL-6, IL-17, IL-23, and IL-22, in the intestinal mucosa during active UC and CD compared to inactive regions and healthy controls [78]. Interestingly, Th17 cells in vivo demonstrate a tendency to transition over time to a Th17/Th1 phenotype characterized by the co-production of IL-17A and IFNγ—or solely to a Th1 phenotype with the cessation of IL-17A production. This phenomenon of Th17 cells is referred to as “plasticity” [79]. A recent study on the plasticity of Th17 cells primarily focused on colitis. Notably, recent data from mouse models of IBD suggest that T-cell plasticity, especially along the Th1/Th17 and Th17-Treg axes, plays a crucial role in regulating intestinal immune responses, pathogenicity, and immune homeostasis. Furthermore, individuals with IBD demonstrate increased numbers of “transdifferentiated” T-cell populations indicative of heightened plasticity [80,81]. However, the exact function of Th17 plasticity and its relevance to IBD in human pathology is largely unknown.

The interplay between different signaling cascades together play a vital role in regulating T cell differentiation, while CD4 T cells work by releasing cytokines, CD8+ T cells are cytotoxic and play a central role in the adaptive immune response, while they have no direct involvement in neutralizing foreign substances. Following T-cell receptor activation and costimulation by antigen-presenting cells, naïve CD4+ T cells undergo differentiation into one of several lineages of T-helper-cell subtypes (Th1, Th2), primarily depending on the cytokines present in the extracellular environment [82]. In the presence of IL-27 and IL-12, naïve CD4+ T cells undergo differentiation into Th1 cells. Th1 cells play a crucial role in host defense against intracellular viral and bacterial pathogens. IL-27 promotes early commitment to the Th1 lineage by activating signal transducer and activator of transcription (STAT1) signaling. This activation induces the expression of the Th1-specific transcription factor, T-bet, and inhibits the expression of the Th2-specific transcription factor, GATA-3 [83]. T-bet serves as the master regulator of Th1 differentiation, promoting the expression of both IL-12 R beta 2 and IFN-gamma, the signature cytokine produced by Th1 cells. The IL-12 R beta 2 dimerizes with IL-12 R beta 1 to form a functional IL-12 receptor complex, rendering the cells responsive to IL-12, which is crucial for Th1 differentiation. IL-12 signaling stimulates STAT4-dependent expression of IFN-gamma and IL-18 R beta. The formation of the IL-18 receptor complex allows IL-18 signaling to further drive IFN-gamma expression through AP-1-dependent transcription.

In addition to activating STAT4, IL-12, along with IFN-gamma, activates STAT1 to maintain T-bet expression and Th1-specific cytokine production. The naïve CD4+ T cell compartment differentiates into effector and regulatory subsets of Th cells in various pathophysiological conditions, contributing to the development of various diseases and modulating tissue inflammation, particularly in autoimmune diseases [84,85,86]. Regulatory T cells (Tregs) are a specialized subset of T lymphocytes that function as suppressive immune cells and inhibit various elements of immune response in vitro and in vivo [87]. The primary function of Tregs, also known as suppressor T cells, is to maintain a balance between cells that promote host defense and those that suppress inflammation involving suppression of successful immune responses and control of self-versus non-self-recognition. Failure of the latter results in autoimmune destruction of host cells and tissue. Like other T cells, T reg cells mature in the thymus, where they are characterized by the variable expression of CD8, CD4, CD25, and FoxP3. Th1 cells represent a lineage of CD4+ effector T cells that play a key role in promoting cell-mediated immune responses. They are essential for host defense against intracellular viral and bacterial pathogens [88]. Th1 cells secrete a specific set of cytokines, including IFN-gamma, IL-2, IL-10, and TNF-alpha/beta. Beyond the cytokines, the expression of certain cell surface receptors serves as distinctive markers for Th1 cells. These include IL-12 R beta 2, IL-27 R alpha/WSX-1, IFN-gamma R2, CCR5, and CXCR3, allowing for the differentiation of Th1 cells from other T-cell subtypes [89]. Th1 cell differentiation and expansion are orchestrated by cytokines that signal through a subset of receptors, including IL-27, IL-12, and IFN-gamma. IL-27 signaling in naive CD4+ T cells induces STAT1-dependent expression of the Th1-specific transcription factor, which, in turn, promotes the expression of IFN-gamma and IL-12 R beta 2. The IL-12 R beta 2 then heterodimerizes with IL-12 R beta 1 to form a functional IL-12 receptor complex, stimulating STAT4-dependent IFN-gamma production and Th1 differentiation. While Th1 cells play a crucial role in clearing intracellular pathogens, an excessive Th1 response has been associated with gut inflammation in IBD [90]. A subset of Th17 cells that exhibit a Th1 signature appears to be specifically implicated in intestinal inflammation in CD and UC. These findings contribute to a deeper understanding of IBD pathogenesis and may offer insights into the effectiveness of anti-IL-12p40/IL-23 therapies and the lack of success with anti-IL-17A treatments, despite the enrichment of Th17 cells [57].

A substantial amount of research has focused on the imbalance between Th17 and Treg cells, both of which differentiate from CD4+ T cells and contribute to inflammatory bowel disease (IBD). Studies have indicated that this imbalance is a contributing factor to IBD. Th17 cells play a role in promoting tissue inflammation, while Treg cells are involved in suppressing autoimmunity in IBD. Therefore, maintaining a balance between Th17 and Treg cells is crucial for proper immune regulation. Various regulatory factors influencing the production and maintenance of these cells, including T-cell receptor (TCR) signaling, costimulatory signals, cytokine signaling, bile acid metabolites, and the intestinal microbiota, play essential roles in regulating the Th17/Treg balance [39,91]. Recent studies have fueled the notion that CD4+ T helper cells play various roles in the initiation and propagation of autoimmune inflammation. Studies have found that Th17 cells infiltrate lesioned tissue from patients with CD and UC, and the amount of the cytokine IL-17 that is specifically secreted by Th17 cells significantly increases [92,93]. Furthermore, in different studies of inflamed tissue samples from patients with UC and CD, the abundance of Th17 cells and the expressions of IL-17A, IL-21, and IL-22 were found to be significantly increased in active IBD patients. These elevated levels correlated with disease activity as well as endoscopic and histological scores. This evidence underscores the crucial role of Th17 cells and Th17-related cytokines in mucosal damage and disease activity in IBD [94,95]. IL-17 plays a critical role in inflammatory and immune mechanisms through which IL-17 is considered a molecular target for the development of novel IL-17A-blocking agents for the treatment of IBD [96]. Several clinical trials have shown multiple factors affecting differentiation and regulation of the Th17/Tregcell balance in IBD. The cytokines involved in regulating the balance of Th17 cells/Treg cells are predominantly inflammatory cytokines. These include transforming growth factor β (TGF-β), IL-2, IL-6, IL-15, IL-18, IL-2, and IL-23. These cytokines play a critical role in influencing the differentiation and maintenance of Th17 and Treg cell populations, thereby impacting the delicate balance between proinflammatory and regulatory responses in the immune system [97,98]. The increased numbers of CD4+CD45RA-FoxP3low cells may lead to an imbalance between Treg and Th17 cells. Notably, this imbalance is primarily localized to the LPC rather than secondary lymphoid tissues [99]. The upregulated secretion of IL-17A and the co-expression of CCR6 in Treg subsets are associated with the imbalance between Treg and Th17 cells in patients with active UC [100]. Imbalance between pathogenic cells and immunosuppressive cells is associated with disease activity of UC, and Tregs are critical for this immune homeostasis. Lack of CD226 expression on FoxP3+Tregs, regardless of TIGIT expression, may play an important role in exhibiting their suppressive function and preventing from disease activity in UC [101].

IL-10 is a crucial cytokine utilized by Foxp3-expressing CD4+ Treg cells to uphold immune tolerance, particularly in the context of maintaining tolerance towards commensal bacteria in the gut. Knockout studies have suggested the function of IL-10 as an essential immunoregulator in the intestinal tract [102]. CD patients react favorably towards treatment with bacteria producing recombinant IL-10, showing the importance of IL-10 for counteracting excessive immunity in the human body. These results have sometimes provided hints into disease pathobiology, [103] a GWAS implicated the IL-12/IL-23 pathway in the development of CD, which supported subsequent clinical trials for drugs targeting the IL-12/IL-23 pathway [104]. Accordingly, cytokines, cytokine receptors, and regulators of signaling are among the most overrepresented class of genes linked to IBD [105]. A subset of IBD-relevant human enteric bacterial species preferentially stimulates bacterial antigen-specific Th1 and Th17 immune responses in this model, independent of luminal and mucosal bacterial concentrations [106]. A significant discovery by Neurath et al. is the association of loss-of-function mutations in the genes encoding IL-10 and the IL-10 receptor with very early-onset (VEO) IBD. Notably, mice that lack the anti-inflammatory cytokines IL-2 or IL-10 exhibit the development of spontaneous colitis, underscoring the crucial role of these cytokines in preventing inflammatory responses in the gut [15].

GWASs mouse models of colitis, and in vitro experiments conducted using murine and human gut tissue, have significantly advanced the understanding of the molecular mechanisms involved in cytokine signaling and their impact on mucosal inflammation [19]; every component of the intestine, from the enteric microbiome to epithelial and immune cells, including antigen-presenting cells (APCs) such as dendritic cells and macrophages, as well as T and B cells, has been implicated in the pathogenesis of IBD [14]. Converging data from GWASs and mouse models have identified more than 240 IBD-associated loci that contain genes that encode cytokines and proteins involved in cytokine signaling and regulate the development and function of Th cell subsets, particularly Th17 pathways and Foxp3-expressing Tregs. An interesting finding was strong evidence linking variants in IL-23R to susceptibility to CD, thus confirming the implication of the IL-23/IL-17 axis in the pathogenesis of disease [14]. Risk alleles in genes associated with Th17 pathways, including CARD9, IL12B, STAT3, RORC, IL23R, JAK2, TYK2, and CCR6, are indeed expected to influence various aspects of Th17 cell biology. These genetic variants may impact Th17 cell generation (e.g., CARD9, IL12B), intracellular events crucial for Th17 lineage commitment and maintenance (e.g., STAT3, RORC, IL23R, JAK2, TYK2), or Th17 cell function (CCR6) [14]. A distinct but tightly regulated T helper cell response is essential for various aspects of host immune function. This includes providing protection against microbial pathogens, maintaining immune tolerance to host tissues and commensal symbionts, resolving inflammation, and fostering the development of durable immune memory in conditions such as CD and UC. Therapies to treat IBD include monoclonal antibodies that either neutralize inflammatory cytokines or their receptors [77]. A study by Friderich et al. summarized cytokine targets in IBD identified by genetics and functional studies [24]. Absolutely, the understanding of IBD pathogenesis has highlighted the fundamental role of cytokines. Consequently, therapeutic approaches have been developed to target these cytokines in order to manage IBD. The following sections provide a description of how the immune system and the cytokine-mediated interactions operate and how these systems can go awry and give rise to innate-derived cytokine initiate inflammation of the gut and progression toward chronic inflammation and IBD.

## 3. Integrins

Immunologically important effector molecules called integrins have recently received much attention. Leukocyte recruitment to inflammation sites is precisely regulated by interactions among endothelial cells, integrins, and the extracellular matrix (ECM) to ensure the proper positioning of immune cells in local environment [107]. During inflammation, integrins play a crucial role in facilitating the movement of white blood cells across the vascular wall. Proinflammatory cytokines contribute to this process by promoting increased binding between integrins and their ligands. This enhanced binding readies the white blood cells to traverse the endothelial surface and ultimately enter the gut mucosa [108]. The integrin family of transmembrane cell adhesion molecules (CAMs) is essential for sensing and adhering to the ECM. Integrins are heterodimeric proteins of the plasma membrane that are critical to cell–cell interactions and to interactions of the cell with ECM proteins [109]. Integrins are considered dimeric broadly distributed cell-surface adhesion receptors of noncovalently associated alpha (α) and beta (β) subunits that engage ECM and couple to intracellular signaling and cytoskeletal complexes. Each integrin contains one α subunit and one β subunit [110]. Eighteen types of α chain and eight types of β subunits associate with each other to form 24 different heterodimers [111]. Integrins mediate leukocyte adhesion and regulate cellular growth, signaling, proliferation, and migration to neighboring cells or ECM. We suggest that they play important roles in apoptosis, tissue repair, as well as in all processes critical to inflammation, infection, and angiogenesis to undertake diverse physiological and pathological pathways. Under physiological conditions, integrins are highly glycosylated and contain a Ca^2+^ or Mg^2+^ ion, which is essential for ligand binding [112], The integrins α4β1, α4β7, αEβ7, and αLβ2 have been implicated as receptors that contribute to leukocyte trafficking. Integrins must first be activated to enhance avidity for their respective ligands serving as cellular keys to direct lymphocyte migration into specific target tissues. For instance, α4β7 is activated by the chemokine CCL25 being expressed in the small intestine, where it interacts with lymphocyte receptors resulting in the binding of integrins to tissue-specific CAMs and the subsequent extravasation and retention of lymphocytes in peripheral tissue, including the gut [113,114]. CD154, which is a costimulatory molecule belonging to the TNF family, has been identified as a new integrin ligand [115], and demonstrated the critical dependence of antibody-secreting cells (ASC), particularly B cells, on the integrin α4β7/MAdCAM-1 interaction for intestinal recruitment. This interaction plays a crucial role in controlling the microbiota during chronic colitis. The research highlighted the importance of α4β7/MAdCAM-1 interactions for B cells/ASC in terms of intestinal recruitment, IgA production, and the maintenance of a homeostatic microbiota, emphasizing the intricate relationships between immune cells, integrins, and gut homeostasis [116].

Leukocyte trafficking to the digestive tract and leukocyte cell-adhesion integrins is clearly recognized to primarily participate in the regulation of inflammation and pathogenesis of IBD [114]. Integrins are expressed in T and B cells, neutrophils, NK cells, monocytes, dendritic cells, macrophages, and platelets, playing a vital role in immune cell adhesion, migration, and interactions within the immune system [117]. There are many different subsets of T cells that modulate adaptive immune responses in the gut. The α4 integrin is found on nearly all lymphocytes, as well as to a lesser extent on monocytes and eosinophils. Typically paired with either a β1 or β7 subunit, α4 integrins primarily interact with endothelial ligands such as vascular cellular adhesion molecule 1 (VCAM-1) and mucosal addressing cellular adhesion molecule (MAdCAM-1). The recruitment of lymphocytes to the gut mucosa involves a complex interplay between integrin α4β7 and MAdCAM-1 [118]. Additionally, the interaction between α4β1 and MAdCAM-1 is involved in an alternative mechanism for recruiting inflammatory T cells to the gut, particularly during chronic intestinal inflammation [119]. The adoptive transfer of α4 null T cells, leading to impaired homing of T cells to inflamed tissues, significantly alleviated chronic colitis in immunodeficient mice [120]. Blocking α4-integrin prevents the immune infiltration of activated T-cell populations that drive IBD [121]. The impact of the α4 integrins on chronic inflammation has also been studied by Binion et al., who highlighted the critical role of α4 integrins in intestinal inflammation and immune cell recruitment by using the immunoblockade of α4 integrin in a cotton-top tamarin model of colitis [122]. Furthermore, several studies demonstrated that endothelial cells extracted from inflamed intestinal mucosa of IBD patients had increased α4-dependent adhesiveness to leukocytes in vitro [123]. A previous GWAS revealed that immune activation of multiple integrin genes (ITGA4, ITGB8, ITGAL, ICAM1) was associated with IBD. Additionally, these four loci were linked to an increased risk of developing IBD [124]. However, the exact contribution of integrins in IBD pathogenesis is up for debate. Despite new information concerning the factors governing lymphocyte migration into the intestinal mucosa and αE integrin expression in healthy and IBD subjects, the need for well-designed studies remains. Integrin-mediated cell adhesion, migration, and signaling are crucial for proper immune system function. In a very interesting study, Keir et al. recently demonstrated the regulation and role of αE integrin and gut-homing integrins in the migration and retention of intestinal lymphocytes in IBD. Their study revealed the upregulation of ICAM1, VCAM-1, and MAdCAM-1 at the gene and protein levels in both ileal and colonic tissues from active IBD patients compared to healthy subjects and/or inactive IBD patients. These findings suggest that cell migration to the gut mucosa may be altered in IBD, and α4β7− and α4β7+ T cells may upregulate αEβ7 in response to TGF-β once within the gut mucosa [125].

## 4. Interleukins

Interleukins are a type of cytokine first thought to be expressed by leukocytes alone, but were later found to be produced by a variety of cells including macrophages, T lymphocytes, mast cells, stromal cells, epithelial cells, and neutrophils [126]. They play essential roles in immunomodulatory functions and the activation and differentiation of immune cells, as well as proliferation, maturation, and adhesion [23], and have pro- and anti-inflammatory properties. The primary function of interleukins is, therefore, to modulate differentiation, and activation during inflammatory and immune responses [127]. Interleukins are categorized into different families based on sequence homology, main functions, and receptors. These families include the IL-1 family, γc family, chemokine family, IL-10 family, IL-6/IL-12 family, and IL-17 family. Additionally, interleukins are classified as Th1-like and Th2-like cytokines based on their immune responses [128] and their dysregulation can lead to IBD. Main interleukins and their function are shown in Table 2.

### 4.1. Proinflammatory Cytokines

Proinflammatory cytokines are produced predominantly by activated macrophages and are involved in the upregulation of inflammatory reactions [27].

#### 4.1.1. Interleukin 1

IL-1α and IL-1β were the first cytokines to be discovered in 1974 by Charles A. Dinarello [23]. The IL-1 family is intricate, featuring ligands with agonist, antagonist, or anti-inflammatory activity and nine receptor chains. Traditionally associated with inflammation and innate immunity, IL-1 has a broader role that extends beyond generic inflammation. IL-1, along with related family members IL-33 and IL-18, plays distinct roles in shaping innate immunity and inflammation in response to various microbial or environmental challenges [130]. Cytokines such as IL-1, IL-6, IL-8, and GM-CSF may play a crucial role in initiating and amplifying the inflammatory response, leading to intestinal injury. There is growing evidence that IL-1 is activated early in the inflammatory cascade. Consequently, IL-1 is considered a primary target for therapeutic intervention in inflammatory diseases, including IBD. Moreover, patients with IBD exhibit a mucosal imbalance between intestinal IL-1 and IL-1ra, indicating that inadequate production of endogenous IL-1ra may contribute to the pathogenesis of chronic gut inflammation [131]. IL-1 contributes to maintaining the equilibrium between immune tolerance to commensal microbiota and the response to intestinal pathogens. The players involved in this process (inflammasomes, IL-1 cytokines, IL-1 receptors, and negative regulators) are expressed by epithelial cells or by leukocytes residing in the mucosa. Several lines of evidence indicate that IL-1 family members, such as IL-1, IL-1Ra, IL-18, and IL-33, possess dual functions depending on the phase of intestinal disease, as well as on their role in initiating vs. sustaining chronic gut inflammation, and finally, on the cell type targeted by the cytokine [130]. IL-1 serves as a key mediator of innate immunity and inflammation, contributing to tissue damage in IBD. In the inflamed mucosa of IBD patients, there is an observed imbalance between IL-1 and its antagonist IL-1Ra, with increased levels of both, but a significantly decreased ratio of IL-1Ra to IL-1 compared to controls. Elevated IL-1β and its receptor are particularly prominent in CD, with a positive correlation between mucosal inflammation severity and IL-1β levels. IL-1β can induce apoptosis in epithelial cells, leading to tissue damage and barrier dysfunction [132]. There was a markedly significant decrease in the IL-1ra/IL-1 ratio in the intestinal mucosa of both CD and UC. This ratio closely correlated with the severity of the disease. Importantly, the decrease in the IL-1ra/IL-1 ratio was specific to IBD, as it was not observed in patients with self-limiting colitis [133]. IL-1 and TNF-α play a well-established inflammatory role in the pathogenesis of IBD. While evidence suggests that these cytokines may have profibrotic effects, the exact impact in vivo is not yet clear. IL-1β and TNF-α have been shown to stimulate the secretion of collagens I/IV, IL-8, monocyte chemoattractant protein-1, and MMP-1 in colonic subepithelial myofibroblasts [134]. The effects of IL-1β and TNF-α extend to human intestinal microvascular endothelial cells, where they contribute to a profibrogenic role in the gut by inducing EndoMT [134]. Adler et al. showed that anti-TNF-α treatment prevents bowel fibrosis in rats with CD [135]. The expression of IL-1 by both myeloid and epithelial cells of the mucosa during IBD was reported, and its levels correlated with the severity of inflammation in experimental models. In contrast, expression of IL-1Ra is significantly decreased in patients with IBD [136]. IL-1 molecules play a major role in host defense mechanisms against microorganisms. During infection with pathogens, tissue damage occurs and microorganisms stimulate IL-1β production and inflammation [136]. In animal models, mice deficient in IL-1Ra (IL-1rn-/-) spontaneously developed intestinal inflammation, offering an effective approach to mimic features associated with IBD. Notably, older IL-1rn-/- mice exhibited a higher inflammatory response compared to younger counterparts. This model provides evidence for the involvement of the imbalance between IL-1 and IL-1Ra in the pathogenesis of IBD [132]. Mak’Anyengo et al. emphasized the NLRP3 inflammasome as a critical checkpoint regulating the IL-1β/IL-18 ratio in the intestine to control immune homeostasis and Th17 immunity. The balance of IL-1β and IL-18 influences the secretion of FLT3L and GM-CSF by T cells, subsequently impacting the differentiation of CD103+ dendritic cells from their precursors. They suggest that pharmacological inhibition of the NLRP3/IL-1β/GM-CSF axis could be a promising approach for the treatment of IBD [137]. Murine models of colitis shown that deoxycholic acid increased the level of IL-1β while reducing the number of tuft cells and upregulating the expression of CD3+ and CD4+ T cells in the intestinal mucosa of mice with dextran sulfate sodium-induced colitis, thereby affecting the intestinal mucosal barrier and intestinal immune functions and aggravating intestinal inflammation in the mouse model [138].

#### 4.1.2. Interleukin 8

IL-8 (CXCL-8) serves as one of the major mediators of the inflammatory response. IL-8 is produced by various types of cells in inflammation. IL-8 is expressed in macrophages and certain other cell types—endothelial cells, leukocytes, and smooth muscle cells—in response to inflammation. The synthesis of IL-8 is strongly stimulated by IL1-β, TNF-α, and bacterial lipopolysaccharides (LPS). IL-8 as a potent chemoattractant is involved in neutrophil activation, the transcription of which is NF-κB-dependent [139]. Two receptors exist for IL-8, CXCR1 and CXCR2 in humans, which belong to the γ subfamily of GTP binding protein (G-protein)-coupled rhodopsin-like 7 transmembrane domain receptors and activate a phosphorylation cascade to trigger chemotaxis and neutrophil activation as part of the inflammatory response [140]. The biological activities of IL-8 resemble those of a related protein, NAP-2 (neutrophil-activating protein-2) and through the binding to its cognate G-protein-coupled CXCR1 and CXC2 chemokine receptors, though the latter has a weaker affinity for IL-8 [141,142].

Evidence suggests that IL-8 drives the inflammatory response in IBD. Dysregulated signaling at the IL-8/CXCR1/2axis may be a possible cause to drive this immunopathology leading IBD formation. Gijsbers and colleagues evaluated the intestinal expression of the CXCR1-binding chemokines IL-8/CXCL8 and GCP-2/CXCL6 and the participation of immunocompetent cells in IBD. They observed downregulated production of IL-8/CXCL8 by leukocytes in CD and selective expression of GCP-2/CXCL6 in inflamed intestinal tissue [143]. IL-8 expression in tissue specimens and mucosal biopsies are altered in UC and in CD [144]. IL-8 levels not only change between subjects with and without IBD, but also between different stages of IBD [145]. For example, Brandt and colleagues described enhanced production of IL-8 in chronic but not in early ileal lesions of CD [146]. One study explored the change and significance of IL-8, IL-4, and IL-10 in the pathogenesis of terminal ileitis. IL-8 can induce the inflammatory reaction in terminal ileitis and chemokine aggregation, and mediates inflammatory reaction by mediating other inflammatory factors in SD rat; as a proinflammatory cytokine, IL-8 can inhibit IL-10; IL-10 and IL-4 can inhibit the inflammatory reaction of terminal ileum [147]. Functional studies have demonstrated that Th17-related effector cytokines induce proinflammatory responses, contributing to the pathogenetic mechanisms of CD. These responses include the recruitment of neutrophils via IL-8 induction; the upregulation of inflammatory mediators such as TNF-α, IL-1β, and IL-6; and the secretion of metalloproteinases by intestinal fibroblasts [148]. Similarly, the expression of the IL-8 gene and the production of IL-8 messenger RNA in IBD are limited to areas exhibiting histological signs of inflammatory activity and mucosal destruction [149]. Furthermore, IL-8 had a significantly increased expression in the colon tissues of the participants with CD, and some genotypes and alleles for the gene polymorphisms rs103284 and rs105432 were significantly higher in the CD group than in the control group. In addition, the disease’s location and behavior were significantly different for participants in the CD group with different genotypes [150]. Studies have demonstrated post-transcriptional regulation and dependency on the NOD2/CARD15 mutations for IL-8 and IL-1β secretion with muramyl dipeptide (MDP). These findings suggest that a signaling defect of innate immunity to MDP may be an essential underlying defect in the pathogenesis of some CD subjects [151]. Interestingly, a meta-analysis indicates that IL-8 rs4073, IL-10 rs1800871, IL-10 rs1800872, IL-10 rs1800896, and IL-18 rs1946518 polymorphisms may influence the predisposition to IBD. Additionally, the IL-18 rs187238 polymorphism may impact the predisposition to CD, but not the predisposition to UC [152]. It is notable that mucosal adherent E. coli is found in IBD and colon cancer. Mucosa-associated E. coli sheds flagellin that elicits epithelial IL-8 release, but this may only become relevant when the mucosal barrier is weakened to expose basolateral TLR5. Adherent and invasive IBD and colon cancer E. coli isolates also elicit a flagellin-independent IL-8 response that may be relevant when the mucosal barrier is intact. The IL-8 release is MAPK-dependent and inhibited by mesalamine [153]. CD14+ MNPs and T cells infiltrate colon in UC. Several findings have established a link between monocyte-like CD163-MNPs, IL-12, IL-1β, and the detection of colonic memory IL-8-producing CD4+ T cells, all of which may contribute to the pathogenesis of UC [56].

#### 4.1.3. Interleukin 17

The IL-17 cytokine family comprises six ligands, IL-17A to IL-17F, and is the key cytokine produced by Th17 cells [154]. IL-17 serves as a pivotal cytokine that connects T-cell activation to neutrophil mobilization and activation [155]. In both UC and CD tissues, there are high levels of IL-17-producing cells and upregulation of RNA transcripts for IL-17A and IL-17F in inflamed guts when compared with healthy controls [156]. One study on children suffering from IBD found higher levels of serum IL-17A than in healthy subjects [157]. IL-17 is also implicated in the local control of barrier integrity and defense against extracellular pathogens, including fungi and bacteria [158]. IL-17, secreted by Th17 cells, plays a multifaceted role. It inhibits the colonization of pathogenic bacteria by targeting intestinal epithelial cells, enhancing the secretion of IgA and antimicrobial peptides. Tregs, on the other hand, inhibit excessive T-cell immunity. Simultaneously, IL-17 induces intestinal epithelial cells to express IL-8, recruiting numerous neutrophils and neutrophil extracellular traps (NETs) to sites of inflammation, positively regulating Th17 cell feedback. IL-21 is involved in autocrine regulation of Th17 cell differentiation. As inflammation progresses, IL-17 can induce fibroblasts to secrete the extracellular matrix, promoting the progression of intestinal fibrosis [159]. Lucaciu and colleagues conducted a study on the use of IL-17 and IL-23 for stratifying IBD patients by disease severity, comparing them with standard inflammatory tests in clinical practice. The results emphasize that IL-23 was particularly effective in differentiating IBD patients with a severe disease phenotype, surpassing the performance of fecal calprotectin. While IL-17 was more elevated in UC patients with severe disease compared to CD, its diagnostic accuracy for disease severity was lower than that of other biomarkers [160]. A study by Zeng et al. strongly supports the evidence that group 3 innate lymphoid cells (ILCs) maintain microenvironmental homeostasis of the gastrointestinal mucosa through the moderate production of IL-22, IL-17, and GM-CSF to protect gut epithelia from microbial invasion in the physiological state. However, they also contribute to the evolution and aggravation of IBD if IL-22 and IL-17, along with IFN-γ, become overexpressed due to dysregulation of ILC3 functions and their transition towards ILC1 in the pathological state [161]. An interesting study by Alexander et al. shows that the potential for broad impacts of E. lenta across diverse disease states is increased due to the effect of this gut bacterium on Th17 cells coupled to the lack of antigen specificity. In contrast, E. lenta can act on Th17 cells post-differentiation and at a distance. These findings raise the potential for effects on Th17 cells in other tissues outside the gut or for synergistic effects with previously described antigen-specific responses [162]. In recent years, IL-17 has been implicated in the pathogenesis of fibrosis, though its specific role in IBD and associated intestinal fibrosis remains controversial. Existing data propose both a proinflammatory and profibrotic action, as well as a protective function of the Th17/IL-17 immune response [163]. A study by Zhang et al. confirmed the involvement of IL-17A in the development of intestinal fibrosis through inducing epithelial–mesenchymal transition [164]. A study by Quing et al. revealed the differentiation of Th17 cells may mediate the abnormal humoral immunity in IgA nephropathy and IBD patients [165]. Interesting work by Fielhause et al. demonstrates that IL-17 inhibitors are safe and highly effective in the treatment of psoriasis and psoriatic arthritis. Adverse effects are rare, including the potential new-onset or exacerbation of IBD, although causality has not been firmly established [166]. A case report by Ju describes a forty-one-year-old Chinese male patient who initially sought treatment for psoriasis, developed severe digestive symptoms following the use of an IL-17 inhibitor, and was subsequently diagnosed with CD. The patient ultimately found relief for both conditions by using an IL-23 antagonist [167]. A study by Moraes et al. suggests that in UC, genes and pathways associated with autophagy, ALPK1, and IL-17 signaling are consistently downregulated, regardless of disease activity. Patients with UC in remission exhibit dysfunctional mechanisms that hinder them from achieving and maintaining true homeostasis [168]. IL-17 was associated with our SERPINE1 gene correlation cluster, and IL-17A was found to induce the expression of Plat, the gene coding tissue plasminogen activator [169].

#### 4.1.4. Interleukin 18

IL-18, belonging to the IL-1 family, enhances IFN-γ production by anti-CD3-stimulated Th1 cells in collaboration with IL-12. Upon stimulation with antigen (Ag) and IL-12 or IL-4, naïve T cells develop into IL-18R expressing Th1, which increase IFN-γ production in response to IL-18 stimulation [170]. Interestingly, IL-18 likewise amplifies expression of IL-22 by ILC3 [171]. IL-18 is a distinctive cytokine that promotes Th1 cell differentiation and triggers the production of IFNγ in Th1 cells and NK cells through NF-kB signaling. Its functions are pleiotropic and depend on the surrounding cytokine milieu. IL-18 production is typically induced by caspase-1 in an inflammasome-dependent manner. Research indicates that increased IL-18 production in the gut epithelium contributes to the breakdown of mucosal barriers [172]. Crawford et al. found that IFN-γ and TNF-α were linked to bovine inflammatory diseases and gut barrier dysfunction in cows; on the contrary, IL-18 stimulates immune cells and induces downstream release of proinflammatory cytokines [47]. Extensive evidence suggests variability in the regulation of the IL-18/IFN-γ axis through caspase-1 among patients with CD. This ex vivo model holds therapeutic relevance for identifying eligible CD patients for new targeted therapies [173]. Importantly, the anti-inflammatory effects of IL-18 were observed in the early stage of DSS-induced colitis, while the proinflammatory effects were observed in the later stages of the disease [174]. Interestingly, continuous suppression of IL-18 using a vaccine improves intestinal inflammation in TNBS-induced murine colitis [175].

#### 4.1.5. Interleukin 12/23

IL-12 and IL-23, heterodimeric cytokines sharing the common p40 subunit, are overproduced in IBD and are believed to play a significant role in mediating or sustaining the inflammatory response in these disorders [176], both of them show pro- and anti-inflammatory features in experiments depending on the circumstances [177].

IL-12 regulates mostly T cells and NK responses, inducing the production of IFN-x03B3γ and the polarization of Th 1, and is an important link between innate resistance and adaptive immunity [178], while IL-23 activates the expansion of Th17 cell programIL-23 binds to the IL-23 receptor (IL-23R) and IL-12Rβ1, excluding IL-12Rβ2. The IL-23 signaling pathway involves two receptor chains and signaling proteins, including Janus kinase 2 (Jak2), tyrosine kinase 2 (Tyk2), STAT3, and STAT4 [179]. The crucial role of IL-23 in the pathogenesis of both CD and UC is well-established. However, the specific downstream effector mechanisms through which IL-23 contributes to chronic IBD remain a subject of ongoing debate and investigation [180]. IL-23 is elevated in the gut of CD subjects and has been demonstrated to restore epithelial barrier integrity and enhance defense against pathogens [181]. A study by Greving et al. highlights a previously unappreciated role for IL-12 in the development of chronic intestinal inflammation and suggests that early in disease, IL-12 is the dominant p40-containing cytokine rather than IL-23 [176]. Clinical trials and mechanisms of action support the effectiveness and safety of IL-12/IL-23 antagonists (ustekinumab, briakinumab) and selective IL-23 inhibition (brazikumab, risankizumab, mirikizumab) in treating CD and UC [182,183]. Similar findings were described in patients with psoriasis [184]. In a study by Bauché et al., it is demonstrated that Foxp3+ Treg cells can suppress intestinal inflammation in an innate model of colitis where the gut injury response is dependent on ILC3 production of IL-22 in the presence of damaging cytokines such as TNF, IL-1β, and IL-23 [185]. Aschenbrenner et al. discovered that monocyte subsets in IBD patients express IL-23, and they identified IL-1α/IL-1β and IL-10 as crucial cytokines controlling IL-23-producing monocytes through auto and paracrine sensing [186]. The sequence of albumin-binding protein variants (e.g., of different REX binders) affects their expression, secretion, and surface display, as well as their conformation in L. lactis. All of the used REX proteins secreted by the L. lactis cells bind human IL-23R and suggest binding to the mouse receptor, making them suitable candidates for further testing in an IBD mouse model [187]. Bhatt et al. indicated that Gpr109a signaling suppresses IL-23 production by dendritic cells [188]. Eftychi’s study in NEMOIEC-KO mice revealed that both IL-12 and IL-23 play crucial yet temporally distinct roles in coordinating chronic intestinal inflammation following epithelial barrier damage [189]. Becker et al. have recently demonstrated high constitutive expression of IL-23 p19/p40 in the terminal ileum. These intriguing findings propose a predisposition of the terminal ileum to undergo chronic inflammatory responses mediated by p40/IL-23, potentially elucidating why Crohn’s disease predominantly manifests clinical symptoms in this specific gastrointestinal region [190].

#### 4.1.6. Interleukin 33

IL-33, a member of the IL-1 family, is released by various tissues and cells in both mice and humans, serving as a potent stimulator for the differentiation and function of Th2 cells and innate lymphoid cells (ILC2) [191]. IL-33, along with its receptor ST2, has the potential to interact with key components of the intestine, such as epithelial cells, the microbiome (comprising commensal and pathogenic bacteria), and mucosal immune cells, including Th2 cells, Tregs, and Th17 cells [192]. IL-33 is naturally expressed in intestinal epithelial cells and acts as an endogenous alarm signal in response to tissue damage. Despite its high expression in inflamed lesions of individuals with IBD, IL-33 exhibits a dual role in animal models of intestinal inflammation, influencing Th2 responses, Th1 inflammation, mucosal regeneration, and fibrosis [130]. IL-33, characterized by its pleiotropic functions, acts both as an extracellular cytokine and a nuclear transcription factor. The interaction between IL-33 and its ST2 receptor plays a crucial role in regulating inflammatory disorders. The IL-33/ST2 axis is a key player in maintaining intestinal homeostasis and is integral to the balance between pro- and anti-inflammatory responses in mucosal defenses [193]. IL-33 exhibits dual functionality, existing in two forms: full-length IL-33 (flIL-33) as an intranuclear gene regulator and mature IL-33 (mIL-33) functioning as an extracellular cytokine released from damaged or necrotic cells. While IL-33 can be rapidly released passively in response to stimuli or cell injury, immune cells can also actively secrete it. Tissues contribute significantly to IL-33 expression, and its levels can rise during inflammation, with tissue-derived IL-33 being crucial for certain inflammatory responses like Th2-induced airway inflammation [194]. ST2, a receptor for IL-33, has two splice variants: the soluble form (sST2) and the membrane-bound form. The soluble form, sST2, serves as a decoy receptor by sequestering free IL-33. In contrast, the membrane-bound form of ST2 activates the MyD88/nuclear factor κB (NF-κB) signaling pathway, enhancing the function of immune cells [193]. The IL-33/ST2 axis plays a crucial role in intestinal fibrosis. During the normal turnover of the intestinal mucosa, both IL-33 and ST2 are expressed in large amounts in the epithelium and stroma. However, uncontrolled expansion of IL-33 can lead to epithelial barrier dysfunction, chronic inflammation, and the development of fibrotic lesions. IL-33 also induces enteric glia to release glial cell-line-derived neurotrophic factor family ligands (GFLs), which contribute to maintaining tight junctions and negatively regulating local inflammatory responses in the intestinal epithelial barrier [194].

Aggeletopolou et al. highlighted an exaggerated activation of NLRs and TLRs in the colonic mucosa, leading to elevated expression of proinflammatory cytokines (IL-6, IL-12, IL-23, and TNF-α) by innate immune cells. Both IL-33 and TLR-associated signaling utilize the MyD88-dependent pathway, activating downstream transcription factors. The combined action of IL-33 and TLRs enhances proinflammatory cytokine responses, disrupting tolerogenic responses against intestinal bacteria. The role of IL-33 in IBD is complex, influenced by the diverse pathophysiology of immune responses classified as Th1- or Th2-related in IBD [193]. The exploration of IL-33/ST2-mediated mechanisms in IBD pathology offers promising therapeutic targets for clinical application in IBD treatment. A study by Bamias et al. indicates that IL-33, associated with Th2 immune responses, also exhibits profibrotic functions [195]. He et al. investigated the expression of full-length IL-33 in the epithelium, leading to the accumulation of IL-33 protein in the nucleus and subsequent secretion. This expression in the epithelium promoted the activation of genes in nearby lamina propria leukocytes and epithelial cells. The gene program activated by IL-33 suggests its involvement in the resolution of the inflammatory response [196]. Lepetuso et. al. found out that the inherent role of endogenous IL-33 within the gut mucosa is the protection, potentially through a mechanism that augments miR-320 expression, inducing epithelial restitution and repair and overall epithelial barrier integrity. In the setting of IBD, particularly during early disease stages, this process may be defective, leading to impaired healing and exacerbation of colitis into a more chronic and sustained inflammatory phenotype [197]. Different results were found by Ngo and colleagues, who suggest that ILC2s facilitate IL-33 mediated tissue protection in DSS colitis, while Tregs seem to play an ILC2-supporting role. This could be due to the fact that ILC2s constitutively express the receptor ST2+ and are therefore able to act immediately upon IL-33 treatment, whereas only a subpopulation of Tregs display ST2 expression [198]. Several studies suggest a functional connection between NOD2 and ILC2s, regulated by the IL-33/ST2 axis, which may mechanistically contribute to early events in the development of Crohn’s disease [199,200].

#### 4.1.7. Interleukin 36

IL-36, a member of the IL-1 superfamily, and its receptor ligands (IL-36R), are overexpressed in both animal colitis models and human IBD patients, exhibiting both pathogenic and protective roles depending on the context. The IL-36 family includes three agonists (IL-36α, IL-36β, and IL-36γ) and two receptor antagonists (IL-36Ra and IL-38). The IL-36 receptor agonists bind to the IL-36R complex, exerting pleiotropic effects during inflammatory settings [201]. IL-36R signaling is activated by intestinal damage, stimulates immune cell infiltration and IECs, and promotes the resolution of intestinal mucosal wounds in vivo [202]. Moreover, IL-36 signaling has been connected to fibrotic conditions affecting the intestine. IL-36R-deficiency has been linked to diminished innate, inflammatory, and Th1 responses in various colitis models. Despite its role in promoting inflammation, IL-36R signaling is also crucial for the resolution of mucosal inflammation and the healing of mucosal wounds, particularly through the promotion of IL-22 expression [130]. In homeostasis, IL-36 cytokine expression is low across various organs, such as the skin, intestines, lungs, and brain. However, during inflammation, IL-36 receptor (IL-36R) agonists are predominantly expressed by keratinocytes, epithelial cells, and inflammatory monocytes/macrophages [201].

Scheibe et al.’s recent study reveals elevated levels of IL-36α and collagen in inflamed tissue from patients with IBD and fibrostenotic CD compared to healthy individuals [201]. Ngo et al.’s findings indicate that ongoing clinical trials are exploring the promising results of monoclonal antibody blockade of IL-36R for IBD treatment. However, further research is needed to understand the precise mechanisms of IL-36R signaling in different phases of IBD and fibrotic complications. Determining whether IL-36R blockade alone is sufficient or if a combination approach with existing therapies is more effective remains an important consideration for future studies. Combining IL-36R blockade with pro-healing or pro-antimicrobial factors could potentially address complications associated with blocking the IL-36/IL-36R axis [201]. Elevated IL-36α expression was observed in tissues from individuals with fibrostenotic Crohn’s disease (CD), accompanied by increased numbers of activated myofibroblasts. Activation of IL-36R in both mouse and human fibroblasts enhanced the expression of genes associated with fibrosis and tissue remodeling, including higher levels of collagen VI compared to control groups [134].

Elevated IL-36A levels were identified in fibrotic intestinal tissues of individuals with IBD compared to controls. IL-36 induced the expression of genes regulating fibrogenesis in fibroblasts. In mice, inhibition or knockout of the IL36R gene resulted in reduced chronic colitis and intestinal fibrosis [203]. Modulating IL-36 may offer potential in preventing or treating fibrotic diseases, providing insights into the mechanistic link between inflammation and fibrosis [204].

#### 4.1.8. Interleukin 38

Interleukin (IL)-38, a newly discovered IL-1 family cytokine, inhibits the synthesis of IL-17 and IL-22 [127], and its deficiency in mice is associated with increased disease activity, weight loss, histological damage, and intestinal permeability [205]. Other studies have shown abnormal expression of IL-38 in the intestine, but not in the circulation of IBD [206]. Fonseca et al. observed significantly higher IL-38 gene expression in active UC compared to active CD [207]. Ohno et al. found that IL-38 was mainly expressed in B cells in the inflamed mucosa of UC patients. IL-38 expression was not detected in other cell types such as T cells, monocytes/macrophages and neutrophils. However, B cells have been reported to secret some kinds of cytokines [208]. This study indicated that IL-38 and IL-36Ra mRNA expressions were increased in the tissue from active and remission IBD patients compared with noninflamed tissues [207]. In patients with active IBD, there was a differential protein overexpression of IL-36α, IL-36β, IL-36γ, IL-36Ra, and IL-38 observed in various immune cells, including intestinal epithelial cells, macrophages, CD8+ T cells, and/or dendritic cells (pDCs), when compared with noninflamed controls [207]. IL-36 cytokines and IL-36Ra could potentially serve as novel therapeutic targets for individuals with gut inflammation.

### 4.2. Anti-Inflammatory Cytokines

Anti-inflammatory cytokines play a crucial role in controlling the proinflammatory cytokine response. Key anti-inflammatory cytokines associated with IBD include IL-1 receptor antagonist, IL-4, IL-6, IL-10, IL-11, and IL-13 [209].

#### 4.2.1. Interleukin 4

IL-4 and IL-13 belong to the Th2 cytokine family, along with IL-3, IL-5, and IL-9. Recent studies have suggested a decrease in IL-4 expression in intestinal biopsies from UC patients. Additionally, IL-4, similar to IL-10, has been demonstrated to suppress the expression of proinflammatory cytokines such as TNFα, IL-6, and IL-1β [129]. Having shown that IL-4 evokes major transcriptome changes in human blood monocyte-derived macrophages and that hM(IL4)s promote epithelial wound repair in an in vitro assay, reduce cytokine-induced epithelial barrier defects, and are beneficial in a murine model of acute colitis [210], IL-4 plays a crucial role in antibody class-switch recombination in B cells, facilitating the secretion of immunoglobulin (Ig)-E [129]. Yang investigated the role of IL-4 in IBD. Despite almost undetectable IL-4 mRNA expression in the intestinal mucosa of patients with CD and UC, the administration of anti-IL-4 led to a significant improvement in oxazolone colitis [134]. Zhou et al. demonstrated that IL-4/IL-13 treatment could inhibit Yes-associated protein (YAP) expression via the PI3K-AKT-β-catenin pathway. They also showed that LPS/IFN-γ stimulation increases YAP protein expression in macrophages [211]. A study by Jayme et al. has shown that IL-4 evokes major transcriptome changes in human blood monocyte-derived macrophages and that hM(IL4)s promote epithelial wound repair in an in vitro assay, reduce cytokine-induced epithelial barrier defects, and are beneficial in a murine model of acute colitis; the study demonstrated the cells’ pro-healing/anti-inflammatory ability and presents this as proof-of-concept support for M(IL4) immunotherapy for IBD [210]. The murine model by Leung showed that bone marrow-derived macrophages treated with IL-4 were shown to block colitis [212]. Daryani et al. suggests that IL-4 polymorphisms might play a role in susceptibility to IBD and clinical features [213].

#### 4.2.2. Interleukin 6

IL-6 is a member of the proinflammatory cytokine family with wide-ranging biological effects on immune cells and on many others. IL-6 is structurally classified as a member of the four-bundle family cytokine [23]. IL-6 is a prototypical cytokine featuring redundant and pleiotropic activity. IL-6 signaling is mediated through its transmembrane-bound IL-6R, mIL-6R, or with a soluble form of IL-6R (sIL-6R), as well as the signal-transducing subunit molecule gp130. Therefore, three modes for IL-6 signaling may occur, in which IL-6 is binding to mIL-6R (classic), to sIL-6R (trans-signaling), or is joined through IL-6R to gp130 on nearby located cells (trans-presentation). These pathways, and the fact that gp130 is ubiquitously expressed, lead to the pleiotropic functions of IL-6 [23,214]. IL-6 signaling is involved in chronic intestinal inflammation in IBD. Many studies have shown that IL-6 is a main inducer of CRP, and IL-6 and sIL-6R levels are positively associated with elevated levels in IBD [215].

Ye et al.’s study revealed up-regulated IL-6 expression in the colonic mucosa of IL-10-deficient mice, suggesting a potential proinflammatory role for IL-6 in the development of spontaneous colitis in IL-10^−/−^ mice [216]. In CD patients, elevated IL-6 production correlates with disease activity, relapse frequency, and inflammation severity. IL-6, upon receptor binding, activates gp130-positive T cells, initiating the translocation of signal transducer and STAT-3, leading to the transcription activation of antiapoptotic genes Bcl-2 and Bcl-xl. Tocilizumab, a humanized anti-IL-6R monoclonal antibody, has emerged as a potential treatment for IBD [217]. IL-6 as a crucial cytokine for Th17 differentiation was upregulated in the inflamed tissue of IBD, as well as colorectal carcinoma [218]. IL-6 plays a dual role influenced by gut microbiota. In the presence of bacterial pathogens and microbiome imbalance, IL-6 contributes to increased inflammatory responses and tissue injuries. Conversely, with a balanced and beneficial microbiota, IL-6 is more involved in maintaining homeostasis and reducing pathological inflammation [219]. Variation in IL-6R gene was found to be associated with risk of CD and UC [220]. Insights into NOD2-driven fibrosis in CD suggest that blocking gp130 may be beneficial for some CD patients, potentially as a complement to anti-TNF therapy [221].

#### 4.2.3. Interleukin 9

IL-9, a cytokine in the IL-2Rγc-chain family, has a pleiotropic function in the immune system. Although many biological functions have been attributed to IL-9, the exact mechanisms in the control of Th9 cells remain poorly defined [222]. In the case of Th9 cells, a combination of TGF-β and IL-4 is required for Th9 induction [223]. IL-9 can function as a positive or negative regulator of immune responses on multiple types of cells [224]. High expression of IL-9 and increased IL-9+ T cells were detected in patients with IBD, especially in those with UC. Moreover, induction of Th9 cells is associated with severity of gut pathology. Furthermore, these findings confirm that IL-9 may be inflammatory or regulatory in human diseases [224]. Some authors have shown that Th9 cells prevent colitis in mouse models. They could additionally demonstrate that Th 9 cells are involved in an experimental model that mimics some aspect of human CD. The discovery that IL-9 deficiency reduced colitis activity in the TNBS (T-cell-mediated colitis induced by the hapten reagent 2,4,6-trinitrobenzenesulfonic acid)-induced colitis model emphasizes the broad relevance of IL-9 in T-cell-dependent intestinal inflammation. This model is associated with mucosal Th1 T-cell responses, unlike the model of ulcerative colitis. This observation illustrates the broad diversity of IL-9 in driving T-cell activation in experimental colitis models [225]. Th9-driven intestinal inflammation is caused by IL-9, which impairs barrier function, resulting in translocation of bacteria into the mucosa [226]. Mice lacking PU.1 in T cells were shielded from colitis, and the administration of an IL-9 antibody suppressed colitis. IL-9, in terms of function, hindered intestinal barrier function and impeded mucosal wound healing in vivo [227]. Further study implicates Th9 cell-derived IL-9 in promoting UC by acting on IECs and impairing intestinal barrier function [228]. Th9 cells and IL-9 were explored for their role in regulating the expression of tight junction proteins crucial for preserving intestinal barrier integrity. Key proteins such as claudins and occludin, vital for maintaining intestinal barrier functions, undergo changes in their expression, contributing to various inflammatory disorders [229]. The role of Th9 cells and IL-9 in human IBD remains uncertain, and while deficiencies in IL-9 may contribute to inflammation control in UC, they could also compromise the protective immunity provided by IL-9 secretion. Contradictory findings in various mouse models of IBD highlight the need for further research to elucidate the actual impact of Th9 cells and IL-9 in human IBD [229]. However, the systemic IL9 level is higher in IBD and corresponds with endoscopic inflammation, suggesting its possible application as a negative marker of mucosal healing in UC [230].

#### 4.2.4. Interleukin 10

IL-10, initially identified as cytokine synthesis inhibitory factor (CSIF), [231] stands out among anti-inflammatory cytokines alongside IL-2, TGF, and the more recently identified IL-25, IL-35, and IL-37 [23]. IL-10 is typically present as a dimer and shares certain structural and functional characteristics with interferon (IFN)-γ [232]. IL-10 is produced by various leukocytes, including macrophages, dendritic cells, neutrophils, NK cells, B-cells, and CD8+ T-cells, with CD4+ T-cells being the primary producers [23]. IL-10 receptors (IL-10Rα and IL-10Rβ) are widely expressed on immune cells, allowing IL-10 to regulate various host defense mechanisms. Finally, IL-10 is a critical immunosuppressive cytokine for regulating intestinal homeostasis, repressing proinflammatory responses and limiting unnecessary tissue disruptions caused by inflammation. G protein-coupled receptor 120 (GPR) is implicated in regulating CD4+ T cell production of IL-10 in the gut to inhibit the development of colitis, which identifies GPR 120 as a potential therapeutic target [233]. Mutations in IL-10R genes’ subunits are associated with hyperinflammatory immune responses in early-onset IBD patients [217].

Children with mutations in Il-10, Il-10RA and Il-10RB genes manifested with severe enterocolitis with perianal lesions and penetrating behavior within first months of life [234]. Buruiana et al. have shown that Il-10 supplementation is not effective in the induction of remission in CD [235]. Mice with Treg cells lacking IL-10 or IL-10Rα are prone to spontaneous colitis, highlighting the role of IL-10 in enabling Treg cells to suppress pathogenic Th17 cell responses in colitis [236]. The anti-inflammatory cytokine IL-10 plays a crucial role in dampening intestinal inflammation and is considered a candidate gene for IBD. Polymorphisms in IL-10R are associated with an increased risk of early childhood-onset UC. Additionally, loss-of-function mutations in IL-10 and IL-10R lead to severe infantile enterocolitis resembling CD, characterized by VEO-IBD [234]. 333-bp deletion in IL10RA was recognized to contribute to four cases of clinically diagnosed VEO-IBD with inconclusive IL10RA mutations. Most importantly, they confirmed that typical clinical manifestations and increased serum levels of IL-10 strongly indicate the existence of IL-10R dysfunction [237]. GWASs have additionally documented common polymorphisms in the IL-10 pathway as risk factors for IBD in adults [238].

#### 4.2.5. Interleukin 11

IL-11, a mesenchymally derived cytokine, supports the growth of certain types of plasmacytoma and hybridoma cells, and acts with IL-3. IL-11 supports megakaryocyte colony formation and maturation, and acts as an autocrine growth factor in megakaryoblast cell lines. In addition, IL-11 stimulates erythrocytopoiesis, enhances antigen-specific antibody responses, induces the synthesis of CRP, inhibits lipoprotein lipase activity and adipocyte differentiation, and promotes neuronal development [239]. The IL-11 gene is therefore a good candidate involved in genetic predisposition to IBD. Klein et al. evaluated the role of IL-11 in IBD, finding decreased expression and a failure to downregulate NFκB expression that could play a crucial role in the pathogenesis of UC [240]. Kiessling et al.’s results indicate functional expression of the IL-11Rα mainly on the epithelial cells within the human colon. IL-11 signals through activation of the Jak1-STAT3 pathway, without inducing anti-inflammatory or proliferative effects in colonic epithelial cells [241].

#### 4.2.6. Interleukin 13

IL-13 and IL-4 belong to the Th2 cytokine family, along with IL-3, IL-5, and IL-9. IL-13 may be critical in regulating inflammatory and immune responses [242]. IL-13 promotes tissue remodeling by influencing crypt stemness and inducing hyperplasia of goblet cells (GC) and tuft cells. This process aims to enhance the clearance of pathogens [191]. IL-13 response appears to be a key pathogenic component of the experimental colitis, as IL-13 neutralization prevents its development [243]. The role of IL-13 in IBD and intestinal fibrosis is well-defined. Perianal fistulizing CD pathogenesis involves increased production of TGF-β, TNF, and IL-13 in the inflammatory infiltrate, inducing epithelial-to-mesenchymal transition and the upregulation of matrix metalloproteinases, leading to tissue remodeling and fistula formation [244]. IL-13 signals through IL-13Rα2, activating the TGF-β1 promoter. Prevention of IL-13Rα2 expression reduces TGF-β1 production in colitis induced by oxazolone or TNBS. IL-13 and its receptor are overexpressed in fibrotic areas in patients with CD [134]. IL-13 and IL-9 disrupt the integrity of the intestinal barrier by upregulating claudin-2 expression in tight junctions and promoting apoptosis of epithelial cells in the barrier [245]. IL-13, a key cytokine in Th2-type immunity, exhibits profibrotic activity in various chronic extraintestinal organs like the liver and lung. This is achieved through IL-13’s capacity to elevate downstream TGFβ, stimulate myofibroblast differentiation, and enhance collagen production [195]. In an experimental model, decreasing central cytokines IL-4 and IL-13 appeared to control gut inflammation through the type 2 proinflammatory pathway of the gut mucosa. However, recent data have suggested that this paradigm is not as straightforward, as a study showed that decreasing IL-13 does not have therapeutic effects [246]. Anti-IL-13 agents hold promise as potential therapeutic strategies for the future management of IBD and other human diseases [247].

#### 4.2.7. Interleukin 22

IL-22 is a member of the IL-10 family and is produced by Th-17 cells, γδ T cells, and newly described innate lymphoid cells (ILCs) [248]. IL-22 is crucial for maintaining intestinal epithelial homeostasis and barrier function, providing protective effects in IBD [237]. IL-22 and IL-18 serve crucial functions in the delicate cytokine balance that determines host defense and inflammation particularly at biological barriers. IL-22 and its receptor are present in the digestive tract, with IL-22 expression linked to inflammation, antimicrobial immunity, and malignancy [248]. IL-22-expressing cells, believed to originate from T cells, are notably elevated in inflamed mucosa of IBD patients. Specifically, there is a significant increase in these cells in active lesions of CD patients, but this response is not observed in infectious colitis [249]. In human colonic subepithelial myofibroblasts, IL-22 triggers the secretion of inflammatory cytokines (IL-6, IL-8, IL-11, and LIF) through NF-κB-, AP-1-, and MAP kinase-dependent pathways. This suggests that T-cell-derived IL-22 plays a significant role in the inflammatory response seen in IBD [249]. IL-22 offers direct support to intestinal epithelial cells by inducing the expression of genes involved in proliferation, wound healing, mucus production, and apoptosis [249], strongly supporting the potential clinical utility of IL-22 as a mucosal healing therapy in IBD [250]. IL-22-dependent activation of STAT3 enhances the transcription of antiapoptotic and pro-proliferative genes such as birc5, pla2g5, smo, myc, mcl1, and regIα. These abilities allow IL-22 to promote mucosal healing by stimulating epithelial cell regeneration with goblet cell restitution [251]. In intestinal disease, IL-22 has been demonstrated to promote barrier repair and a return to homeostasis in certain contexts. However, IL-22 has also been shown to promote inflammation and accelerate tumor growth in preclinical models [252]. Thus, the authors suggest that IL-22 regulates the mucus layer by downregulating goblet cells and secreted mucin-2 while upregulating membrane-bound mucins [250]. The study by Powell et al. suggests that the IL-22/endoplasmic reticulum (ER) stress axis may be particularly crucial in chronic inflammation, where other proinflammatory and proapoptotic mediators, such as IL-17A and TNFα, are excessively and persistently produced [253]. IL-22 also plays a role in an IL-18-dependent epithelial response circuit that reinforces intestinal host defense [254].

Complex cytokine network is shown in Figure 1.

## 5. Tumor Necrosis Factor Alpha (TNF-α)

Two independent lines of investigation, including experimental and clinical trials, have strongly implicated TNF-α in IBD pathogenesis. TNF-α has a pleiotropic effect, which is produced by many types of immune and nonimmune cells and is widely implicated in IBD pathogenesis [255]. However, its production can be regulated at multiple levels [256]. TNFα is responsible for the regulation of immune cells and signaling events within cells. Sethi JK et al. provided an overview of TNF in the context of metabolic inflammation or metaflammation, its discovery as a metabolic messenger, its sites and mechanisms of action, and some critical considerations for future research. Additional observation suggests that TNF has been implicated in promoting metabolic inflammation and its discovery as a metabolic messenger [256].

### 5.1. TNF-α Signaling Pathway

TNF alpha exerts many of its effects by binding, as a trimer, to either a 55 kDa cell membrane receptor termed (TNFR-1/TNFR-55) or a 75 kDa cell membrane tissue-restricted receptor (TNFR-1/TNFR-55). TNF, primarily produced by T and innate immune cells, is a potent proinflammatory cytokine, along with others like IFN-γ and IL-17 produced by Th1 and Th17 cells [257]. Optimal regulation of TNF signaling is essential to maintain tissue homeostasis and prevent inflammatory pathology [258]. Upon binding of tumor necrosis factor α (TNFα) to cell surface receptors, multiple signal transduction pathways are activated. These include three groups of mitogen-activated protein (MAP) kinases: extracellular-signal-regulated kinases (ERKs), cJun NH2-terminal kinases (JNKs), and p38 MAP kinases. These pathways initiate a secondary response by enhancing the expression of various inflammatory cytokines, including TNFα, thereby amplifying the biological activity of TNFα. In essence, MAP kinases play a crucial role both upstream and downstream of TNFα receptor signaling [259]. Dependent on the cellular context, TNF-α promotes the regulation of immune homeostasis, the induction of inflammation and apoptotic cell death, and host defense, and is able to inhibit tumorigenesis and viral replication. TNF α effects are mediated through binding and activation, as a trimer, to two distinct cell membrane receptors TNFR1 (TNFRSF1A, CD120a, p55) and TNFRII (TNFRSF1B, CD120b, p75), which initiate signal transduction pathways. Both these receptors are members of the TNF receptor superfamily. Indeed, TNFR1 is widely expressed on most cells and is considered the primary mediator of the cytotoxic effects induced by tumor necrosis factor alpha (TNFα) [260]. Certainly, Alam MS et al. demonstrated the pivotal role of TNF in inflammation through its signaling via T cell TNFR2. Their findings revealed that TNF influences the inflammatory response by utilizing its less-studied receptor, TNFR2, to promote the differentiation of T cells into inflammatory Th17 cells and enhance the production of inflammatory cytokines by Th1 cells. Inhibition of TNFR2 signaling led to reduced disease severity in mouse models of multiple sclerosis and colitis [261]. Certainly, Th17 cells exhibit plasticity and play a significant role in colitis pathogenesis through dual mechanisms. They can transition directly to Th1-like cells, and they also support the development of classic Th1 cells with inflammatory properties [262]. The superfamily includes FAS, CD40, CD27, and RANK. These pathways subsequently lead to activation of NF-kB or MAPK signaling pathways, thereby controlling expression of cytokines, immune receptors, growth factors and cell cycle genes, which in turn regulate inflammation, survival, cell migration, proliferation, and differentiation. Another pathway that TNF-alpha can activate utilizes the death domain of TNF RI to induce apoptosis. TNF-alpha protein is translated as a type II transmembrane protein containing an N-terminal transmembrane domain. The soluble cytokine is released from its cell-anchoring TM domain by proteolytic processing by metalloproteases. Indeed, TNF-alpha promotes the inflammatory response primarily through the activation of the transcription factor Nuclear Factor-kappa B (NF-κB) signaling. Dysregulation of TNF-α production has been implicated in various human diseases [263]. This whole process of TNFR1-induced NF-κB signaling and cell death depends on protein–protein interactions and post-translational modifications. Van Quickelberghe, E et al. described a protein–protein interaction map of the TNF-induced NF-κB signal transduction pathway. Their dataset revealed dynamic interactions in TNFR1-induced NF-κB signaling and identifies both known as well as novel interactors that may help to further unravel the molecular mechanisms steering TNF-induced inflammatory signaling and pathology [264]. Finally, it is therefore not surprising that common proinflammatory molecular players, including TNF and the inappropriate or excessive activation of TNF-α signaling networks, have been implicated in promoting the pathogenesis of chronic inflammation. Mechanistically, it is worth highlighting that dysregulated TNF expression has been linked to the development of pathological complications by acting on tissues, particularly in autoimmune diseases and IBD [265,266].

### 5.2. TNF-α-Induced Protein 8-like 2 (TNFAIP8L2, TIPE2)

TNF-α-induced protein 8-like 2 (TNFAIP8L2, TIPE2) is a newly discovered negative immunoregulatory domain that plays a vital role in regulating inflammatory and cellular immune responses, and is an essential negative regulator of both innate and adaptive immunity by maintaining immune homeostasis [267,268]. TIPE2 appeared to be a critical immunoregulatory molecule involved in the immunosuppressive function of CD4(+)CD25(+) T reg cells [269]. Sun et al. reported that depletion of TIPE2 was related to fatal inflammatory diseases in TIPE2-deficient mice. Additionally, it has been identified that TIPE2 inhibits the activation of NF-κB and AP-1, which are involved in inflammatory and antigen-specific immune responses [270]. Interestingly, although caspase 8 is one of the target molecules of TIPE2 [270], and TIPE2 has been confirmed to inhibit caspase-mediated apoptosis [271]. Recently, Oho et al. was first to demonstrate that TGF-β-activated kinase 1 (TAK1) was another novel target of TIPE2. TIPE2 interacts with TAK1-mediated signals, a crucial regulatory molecule of inflammatory and immune signals, and consequently acts as a powerful negative regulator of these TAK1 signals [272]. Consistently, TAK1 is a member of the MAPK kinase (MAPKKK) family and has been implicated in the regulation of a wide range of physiological and pathological processes. TAK1 functions through assembling with its binding partners TAK1-binding proteins (TAB1, TAB2, and TAB3) (a central signalosome in inflammatory responses) and can be activated by a variety of stimuli such as TNFα, IL-1β, and TLR ligands. Thus, they play essential roles in the activation of NF-κB and MAPKs [273]. Other concerns include TIPE2 with respect to dendritic cells (DCs). TIPE2-deficient DCs are more immature under homeostatic conditions and consequently promote the induction of peripheral T reg cells in the gut mucosa. However, the underlying mechanism by which TIPE2 affects the immune function of DCs is not yet understood. Mechanistic studies revealed that TIPE2 promotes the expression of DC maturation markers CD80 and CD86 through the activation of PI3K-PKCδ-MAPK signaling pathway during the differentiation of DCs. Going forward, these findings suggest that, in addition to acting as a negative regulator of pathogen-induced immune response, TIPE2 in DCs is also capable of promoting immune response under homeostatic condition through the suppression of peripheral tolerance [274]. By genomic sequence analysis, Sun et al. mapped the human TNFAIP8L2 gene to chromosome 1q21.2-q21.3 and the mouse Tnfaip8l2 gene to chromosome 3F1-F3 [270].

### 5.3. TNF-like Cytokine 1A (TL1A)

TL1A (TNF-like cytokine 1A) is a member of the TNF superfamily (TNFSF15) and signals through association with death domain receptor 3 (DR3). In inflamed intestinal tissues, TL1A and DR3 are significantly upregulated, suggesting their pathogenic importance in inflammatory bowel disease (IBD). TL1A/DR3 induce costimulatory signals to activated lymphocytes, impacting major effector pathways and inducing mucosal upregulation of Th1, Th2, and Th17 factors. Treg lymphocytes, expressing DR3, also respond to TL1A stimulation. Genetic studies and therapeutic blockade with anti-TL1A antibodies support the critical involvement of TL1A/DR3 pathways in IBD pathogenesis, including chronic mucosal inflammation and fibrosis reversal. GWASs have identified IBD-specific polymorphisms in the TNFSF15 gene, serving as poor prognostic factors. TL1A blockade in mice has shown promise in reversing established intestinal fibrosis. TL1A/DR3 signaling is implicated in extraintestinal inflammatory conditions associated with IBD. This evidence positions TL1A/DR3 as a potential target for personalized IBD therapy [275]. Polymorphisms in the TNF family member TL1A gene are associated with the development of IBD, and increased serum concentrations of TL1A have been demonstrated in patients with various chronic inflammatory disorders [276]. In mouse models, TL1A has been shown to be a costimulating cytokine that optimizes the Th1 and Th17 responses, inducing inflammation [77].

## 6. Cytokine Targeting Therapies

### 6.1. Anti-Lymphocyte-Trafficking Agents

Leukocyte trafficking to the digestive tract is considered to play an important role in the pathogenesis of IBD. Integrins, expressed on the cell surface, play a crucial role in various inflammation-related processes, making them appealing targets for the development of IBD therapies [4,7]. Anti-integrins exert their effects on various targets and modulate different physiological mechanisms. They block the efflux of immune cells from the vascular compartment into GI mucosal tissues by occupying ligand-binding sites. Anti-integrin therapy inhibits the interaction of integrins on the surface of leukocytes and endothelial CAMs, preventing cells from interacting with the intestinal mucosa. Recently, the blockade of the gut-tropic integrin α4β7 and its subunits has been explored as a therapeutic target in IBD [277]. 

Treatment with anti-inflammatory agents is ineffective in preventing the development of fibrosis in IBD, which is a consequence of chronic inflammation [278]. Cells are forced to undergo responsive changes that influence remodeling during physiological and pathological events. Integrins recognize these changes and trigger a series of cellular responses, forming a physical connection between the interior and the outside of the cell [279]. The interaction between the aberrant release of ECM components and the communication of αv, β5, αvβ8, and αvβ3, which are the major integrin isoforms, and their main function is to activate TGF-β pathways that have been implicated in mediating fibrosis on IBD [280]. Elevated matrix stiffness induces the activation of colonic myofibroblasts, promoting a fibrogenic phenotype and self-propagation of fibrosis [281]. The upregulation of genes associated with inflammatory and fibrogenic remodeling indicates the coexistence of fibrosis and inflammation in Crohn’s disease strictures.

Several integrin inhibitors have been developed, with only a small subset undergoing clinical evaluation. Currently, seven drugs targeting four integrins (αIIbβ3, α4β7, α4β1, and αLβ2) have been successfully marketed, including abciximab, eptifibatide, tirofiban, natalizumab, vedolizumab, lifitegrast, and carotegrast. Notably, vedolizumab and natalizumab, acting on leukocyte integrins α4β7 and α4β1, have demonstrated effectiveness in Crohn’s disease, ulcerative colitis, and multiple sclerosis [282].

Targeting the α4β7 integrin through the antagonist vedolizumab (VDZ) is one of the current therapeutic approaches against inflammatory bowel disease (IBD) [283]. VDZ (monoclonal antibody to the anti-α4β7 ligand-binding site, blocks VCAM-1 binding) is approved for treating patients with UC and CD, and has been shown to be effective in both induction and maintenance therapy. Unlike natalizumab, vedolizumab does not bind α4β1. The efficacy of vedolizumab and a favorable long-term safety profile with few systemic adverse effects has been verified in the several trials (GEMINI 1, GEMINI 2, GEMINI 3, GEMINI LTS, C13002, C13004) [284,285,286]. A randomized placebo-controlled trial of a humanized monoclonal antibody to α4 integrin has proved the effectiveness in patients with active CD [287]. Additionally, findings from the OBSERV-IBD real-world cohort study indicated that vedolizumab can sustain steroid-free clinical remission in both UC and CD patients up to week 162. However, there was a 10% annual rate of loss of response, leading to the discontinuation of vedolizumab [288].

Natalizumab (monoclonal antibody to anti-α4β1 binding site, inhibits the interaction of VCAM -1 with MAdCAM-1 binding) is the first α4 integrin antagonist in a new class of selective adhesion-molecule inhibitors. Natalizumab may also modulate ongoing inflammatory reactions by inhibiting the binding of α4-positive leukocytes with fibronectin and osteopontin. It was the first drug approved for the treatment of relapsing–remitting CD, but its use is limited because of its risk of progressive multifocal leukoencephalopathy, a rare but often fatal neurologic disease [277,289,290,291]. The ENCORE study demonstrated efficacy of natalizumab in CD response and remission, [292]. The ENACT—2 study examined natalizumab as continuous therapy [293].

The benefits of natalizumab in patients with CD disclosed a strong motivation to develop more specific agents targeting α4 integrins in the digestive tract. This was accomplished by targeting α4β7, β7 and MAdCAM-1. VCAM-1 and MAdCAM-1 are upregulated on intestinal endothelium in CD. The efficacy of natalizumab in CD is very likely due to the blockade of leukocyte adhesion factors α4β1 and α4β7 in tandem [294]. Considering this potential concern, various preclinical and clinical studies have presented evidence supporting the anti-inflammatory effects of α4β7 blockade in both experimental intestinal inflammation and clinical trials. The landscape of integrin-based therapeutic drugs or imaging agents in clinical studies is diverse, encompassing approximately 90 types, including small molecules, antibodies, synthetic mimic peptides, antibody–drug conjugates (ADCs), chimeric antigen receptor (CAR) T-cell therapy, imaging agents, and more [7,278,282].

Numerous additional anti-integrin drugs are currently in various stages of development. One such example is etrolizumab, a gut-targeted humanized IgG1 monoclonal antibody that specifically targets the β7 subunit of the α4β7 and αEβ7 integrins. This antibody impedes leukocyte trafficking through α4β7 and inhibits cell adhesion via αEβ7, disrupting their interaction with ligands MAdCAM-1 and E-cadherin, respectively. The α4β7 and αEβ7 integrins play pivotal roles as trafficking molecules, guiding leukocytes to inflammatory sites in the gut [290]. Consequently, the dual targeting of α4β7 and αEβ7 with etrolizumab is anticipated to modulate intestinal inflammation by reducing both leukocyte recruitment into the gastrointestinal mucosa and cell retention within the intraepithelial space of the gut [295,296]. A robust phase 3 randomized clinical trial, enrolling more than 3000 patients, evaluating the safety and the efficacy of etrolizumab in the induction and maintenance of patients with IBD, has been completed, but the results are still not fully available [297]. The etrolizumab phase 3 clinical program comprises six randomized controlled trials (RCTs)—HIBISCUS I and II, GARDENIA, LAUREL, HICKORY (for UC), and BERGAMOT (for Crohn’s disease)—along with two open-label extension trials, COTTONWOOD (for UC) and JUNIPER (for Crohn’s disease). These trials are evaluating patients with moderately to severely active UC or CD [298,299].

Etrolizumab, an anti-α4β7/αEβ7-integrin, has demonstrated effectiveness in the maintenance therapy of CD [300]. In recent updates from phase 3 studies of etrolizumab in patients with moderate-to-severe UC, etrolizumab effectively induced remission compared to placebo but did not meet its primary endpoint as maintenance therapy. There were no major safety issues reported in any of the phase 3 studies to date [298,301]. Numerous new anti-integrin therapies are currently being investigated in various phases of clinical trials. These include abrilumab (anti-α4β7 IgG2), PN-943 (orally administered and gut-restricted α4β7 antagonist peptide), AJM300 (orally active small molecule inhibitor of α4), PTG-100 (anti-α4β1 integrin), ontamalimab (anti-MAdCAM-1 IgG), and carotegrast Methyl (AJM300). These therapies represent a diverse range of approaches in targeting integrins for potential treatment of IBD [277,302].

Abrilumab (AMG181/MEDI7183) is a human monoclonal antibody targeting the α4β7 integrin, sharing a similar mechanism of action with vedolizumab. It has shown positive clinical outcomes in ulcerative colitis (UC) but not in Crohn’s disease (CD) during phase 2 trials, maintaining a favorable safety profile across studies. As of now, phase 3 trials for abrilumab have not been registered. Among oral small molecules targeting integrins for IBD therapy, AJM300 (anti-α4β7 and α4β1 integrins) and PTG-100 (anti-α4β1 integrin) are considered promising. PTG-100 recently demonstrated proof-of-concept efficacy in the PROPEL phase 2a trial involving UC patients. PN-943 has shown greater effectiveness for inducing remission in UC compared to PTG-100. It is an earlier-generation orally administered and gut-restricted α4β7 antagonist cysteine knot peptide developed by the same company, exhibiting a preference for binding activated α4β7 on T cells [303]. Carotegrast methyl is an orally active small molecule inhibitor targeting the α4 integrin. It has shown activity against both α4β7 and α4β1 integrins.

Ontamalimab (PF-00547659) is a human monoclonal antibody that targets MAdCAM-1. Unlike other anti-integrins that bind to circulating leukocytes, ontamalimab acts directly on endothelial cells in the intestine with a highly selective mechanism of action. However, its efficacy has shown mixed results, with encouraging data in ulcerative colitis (UC) such as in the TU-RANDOT study, but disappointing efficacy in Crohn’s disease (CD) as observed in the OPERA trial [304,305]. Preliminary data suggest that several of these new anti-integrin drugs may be more effective in ulcerative colitis (UC) than in Crohn’s disease (CD). However, the majority of phase 3 clinical trials are still ongoing, and complete results are not yet available. These promising anti-integrin agents are in advanced stages of development [302].

### 6.2. IL-12/IL-23 Inhibitors

Targeting the IL-23 axis is already being applied in clinical practice. Inhibitors of IL-12/23 (p19/p40) as well as specific blockers of IL-23 have been explored as potential options for medical therapy in patients with IBD [306]. Indeed, the IL-23 receptor is minimally expressed on naïve T cells, suggesting that its effects are likely exerted on effector T cells located at mucosal sites, in addition to its impact on innate and innate-like lymphocytes [307]. Anti-IL-12p40 agents rendered their anti-inflammatory effect primarily via inhibition of IL-23. This is due to the shared subunit of IL-12/IL-23. Targeting IL-12 selectively was found to be ineffective. Coblockade of IL-12 and IL-23 via targeting of p40, however, have scientific rationale and was found to be effective [308]. The increased understanding of the proinflammatory effects mediated by IL-12 and IL-23 has resulted in the development of monoclonal antibodies targeting a subunit common to IL-12 and IL-23 (p40), such as ustekinumab and briakinumab, or the IL-23-specific subunit (p19), which is targeted by risankizumab, guselkumab, brazikumab, and mirikizumab. Accumulating evidence has shown the efficacy of ustekinumab, a human IgG1 monoclonal antibody targeting the shared p40 subunit of IL-12/23 for the treatment of CD and UC patients both in randomized clinical trials and real-life experiences [309,310,311,312,313,314,315,316,317]. Ustekinumab is a human IgG1 monoclonal antibody currently approved for the treatment of psoriasis, psoriatic arthritis, and moderate–severe CD and UC [318,319,320]. In a substantial real-world Israeli cohort study involving non-naïve-to-biological-treatment CD patients, ustekinumab demonstrated effectiveness and safety in inducing clinical remission. The treatment led to a significant reduction in the number of patients requiring steroids [321]. Furthermore, 16 week data from the UNIFI Trial demonstrated early symptomatic improvement after ustekinumab, in patients with UC during the initial 16 weeks of treatment [322]. Ustekinumab has demonstrated effectiveness in inducing and maintaining clinical, endoscopic, and histologic remission in moderate-to-severe UC in both phase 3 clinical trials and real-world studies. The favorable risk–benefit ratio, efficacy on extra-intestinal manifestations, and effectiveness in patients who have failed other biologics position ustekinumab as an ideal candidate for first-, second-, or third-line therapy in UC [323]. For patients on ustekinumab therapy, the decision on whether to continue without endoscopic response evaluation may be considered in those with a decrease in fecal calprotectin (FC) levels of ≥500 µg/g at week 8. However, in patients without a decrease in FC level, the decision on continuing ustekinumab therapy or optimizing therapy needs reconsideration. In all cases, endoscopic response evaluation during induction therapy remains essential for guiding therapeutic decisions [324]. In the UniStar LTE phase 1 multicenter study spanning 16 and 268 weeks, the pharmacokinetics, safety/tolerability, and efficacy of ustekinumab in the pediatric CD population were assessed. The observed parameters were comparable to those observed in adults during the initial 16 weeks of the study [325] and were generally consistent with adults for as long as 4 years of treatment. These results suggest a different dosing regimen may be required for patients <40 kg from that employed in this study; additional pharmacokinetic analyses may be needed in this population [325]. Ustekinumab appears to be a potentially effective and safe treatment option for pediatric and adolescent CD patients, as well as those with CD disease-like IBD, especially in cases of nonresponse or adverse reactions to anti-TNF agents [326]. Briakinumab, a monoclonal antibody that selectively modulates the IL-23 and Th17 cell pathways, is another therapeutic agent. It blocks the p40 subunit of both IL-12 and IL-23, similar to ustekinumab. Briakinumab has been studied in CD through two phase 2 trials [327]. The pediatric trial on the use of ustekinumab, as indicated by Chavaness et al., provides evidence for favorable response and remission rates in cases of Crohn’s disease (CD) refractory to conventional TNF blockade. Nearly 40% of the cohort achieved clinical remission at 12 months, accompanied by a significant drop in the Pediatric Crohn’s Disease Activity Index (aPCDAI) of nearly 20 points at 12 months [328]. To date, several clinical trials evaluating the clinical rationale of selective IL-23p19 antagonists with promising results from phase II of brazikumab, guselkumab, tildrakizumab, mirikizumab, and risankizumab in moderate-to-severe CD and UC are under investigation (ClinicalTrials.gov identifier (NCT number): NCT05197049, gov, Number: NCT03466411, Galaxi 1, ClinicalTrials.gov Identifier: NCT02589665), but additional studies are warranted [307,329,330,331,332]. A number of IL-23p19-specific antibodies are at advanced stages in clinical trial programs and have now entered phase III studies for induction and maintenance therapy for IBD [306,330,332,333]. Two new studies report randomized controlled trials of risankizumab for CD. The first reports demonstrating the therapeutic effect of IL-23-specific inhibition in phase 3 trials for individuals with Crohn’s disease (CD) are presented in the ADVANCE and MOTIVATE trials. Geert D’Haens and colleagues compare the efficacy and safety of risankizumab with placebo during the induction period in these phase 3 trials [334]. In the second set of reports, Marc Ferrante and colleagues present the results of the FORTIFY phase 3 trial, which compares the efficacy and safety of risankizumab with placebo during the maintenance period [335]. Overall, the classes of anti–IL-12/IL-23 agents and selective IL-23 inhibitors seem to be effective alternatives in subjects nonresponding to anti-TNF-α agents, especially in secondary nonresponders. Further studies must resolve variable efficacy of more specific selective IL-23 inhibition and apparent superiority over IL-12/IL-23 effects of ustekinumab. Additionally, the immunogenicity and minimal adverse events associated with anti-IL-12 and/or IL-23 therapies seem to be very low [336,337]. Wang et al. demonstrated the pathophysiological mechanisms associated with and potentially mediating the response of risankizumab and upadacitinib for IBD patients who inadequately responded to anti-TNF-α treatment. The study integrated eight tissue transcriptomic datasets from IBD patients treated with anti-TNF-α therapies, along with single-cell RNAseq data from UC, to identify TNF-IR mechanisms. RNAseq colon tissue data from clinical studies of TNF-IR CD patients treated with upadacitinib or risankizumab were used to identify TNF-IR mechanisms that were favorably modified by upadacitinib and risankizumab. The findings suggest that upadacitinib and risankizumab may block pathways that remain active in patients with IBD who are TNF-IRs, potentially accounting for their clinical response among TNF-IR IBD patients. The authors suggest that JAK 1 inhibitor upadacitinib and IL-23 risankizumab affect TNF inadequate nonresponders’ upregulated mechanisms, which may account for their clinical response among TNF inadequate nonresponders. Collectively, the study identified seven TNF-IR upregulated modules related to innate/adaptive immune responses, interferon signaling, and tissue remodeling, and six TNF-inadequate responders upregulated cell types related to inflammatory fibroblasts and monocytes, postcapillary venules, macrophages, dendritic cells, and cycling B cells. Upadacitinib was associated with a substantial decrease in the expression of most TNF-inadequate responders upregulated modules in JAK1 responders. In contrast, there was no change in these modules among TNF-inadequate responders treated with a placebo or among JAK1 inadequate responders. Additionally, four of the six TNF-inadequate responders upregulated cell types were significantly decreased after upadacitinib administration in JAK1 responders but not among subjects treated with a placebo or among JAK1 inadequate responders. Similar findings were observed in colon biopsy samples from TNF-inadequate responders treated with risankizumab. The results suggest that upadacitinib and risankizumab may block pathways that remain active in patients with IBD who are TNF-IRs, potentially explaining their clinical response among TNF-IR IBD patients. Understanding these mechanisms may aid in the development of new IBD treatment strategies [338]. The expansion of apoptosis-resistant intestinal TNFR2+IL23R+ T cells is associated with resistance to anti-TNF therapy in Crohn’s disease (CD). Responders to anti-TNF therapy showed significantly higher expression of TNF receptor 2 (TNFR2), but not IL23R on T cells compared to nonresponders before anti-TNF therapy. During anti-TNF therapy, there was a significant upregulation of mucosal IL-23p19, IL23R, and IL-17A in anti-TNF nonresponders but not in responders. These findings highlight IL-23 as a potential molecular target in CD patients refractory to anti-TNF therapy [339]. Third generation, currently in development which targets the IL-23 pathway, includes synthetic small molecules such as oral IL-23R antagonists, small molecule inhibitors of signaling molecules activated downstream of the IL-23R, the TYK2 inhibitors deucravacitinib, PF-06826647 and type I interferons which may have advantages over biologics; however, it will be important to determine if this is true in IBD patients [307].

### 6.3. TNF-α Targeting Drugs

TNF-targeting agents have become a cornerstone of IBD management as first-line biologicals for IBD, in both step-up and top-down approaches. The combined action of two mechanisms (the induction of T-cell apoptosis and the Fc-receptor-dependent promotion of reparative macrophages) may explain inhibition of inflammation and mucosal healing. Nevertheless, despite these advancements, in many patients, the effectiveness of anti-TNF therapy remains suboptimal (inadequate responders-IR) due to inadequate primary or secondary nonresponse to anti TNF after initial success, rather than adverse effects [340]. After the failure of the initial anti-TNF-α treatment in an inflammatory bowel disease (IBD) patient, the next step involves choosing between a second anti-TNF-α or a drug with a different mechanism of action. The decision is based on individual patient factors and specific disease considerations [341]. At present, there is no established guidance for determining the most suitable second-line therapy following the failure of anti-TNF treatment [342]. The reduced effectiveness of second-line biological therapy in patients previously exposed to anti-TNF has direct clinical implications, as these individuals are at a greater risk for unfavorable outcomes [343]. It has also been shown that up to 40% of IBD patients show primary or secondary nonresponse [344]. Loss of efficacy to anti-TNF occurs when patients initially respond to anti-TNF treatment but subsequently and progressively lose this response. Reports indicate that up to 50% of individuals experience a loss of response within the first year after induction, with an annual rate of 5–20% [345]. Therapy resistance to anti-TNF can be attributed to various factors, including pharmacokinetic or mechanistic issues. Factors such as low trough serum drug concentrations and the development of antidrug antibodies (ADAbs) can result in suboptimal drug concentrations or a reduction in TNF-binding capacity, contributing to treatment resistance [346]. Identifying patients at high risk of immunogenicity is crucial for personalized medicine when selecting TNF-α antagonists for immune-mediated inflammatory diseases (IMIDs). Solitano et al. conducted a meta-analysis of 13 studies and found that variants in HLA-DQA1∗05 are associated with an increased risk of immunogenicity and secondary loss of response (LOR) in patients with IMIDs treated with TNF-α antagonists. However, the positive and negative predictive values are moderate, emphasizing the need for individualized decisions, including the consideration of concomitant use of immunomodulators to prevent immunogenicity [347]. In a retrospective cohort study by Fuentes-Valenzuela et al., involving 112 patients initiating anti-TNF therapy under proactive therapeutic drug monitoring (PTDM), it was observed that HLA-DQA1*05 carriers did not exhibit lower drug persistence or remission rates. This suggests that PTDM may overcome the anticipated reduction in treatment survival expected in HLA-DQA1*05 carriers. In adult patients with PTDM, a positive HLA-DQA1*05 genotype does not appear to be associated with a higher risk of treatment cessation or worse clinical outcomes [348].

Recent studies have linked the risk of antibody development against anti-TNF agents to the HLA profile of subjects. The authors found a GWASs between HLA-DQA1*05 and antibody formation against anti-TNF agents in addition to infliximab and adalimumab LOR and treatment discontinuation. A randomized controlled biomarker trial (ClinicalTrials.gov ID: NCT03088449) is deemed necessary to investigate the potential improvement in patient outcomes by incorporating pretreatment testing for HLA-DQA1*05. This approach aims to assist physicians in selecting anti-TNF and combination therapies, considering the significant correlation between HLA-DQA1*05 carriage and the development of antibodies against anti-TNF agents [349,350,351]. Despite maintaining adequate anti-TNF antibody serum trough levels and lacking detectable neutralizing antibodies, a substantial subgroup of individuals experience secondary nonresponse to anti-TNF. This suggests that, beyond pharmacodynamic factors, alternative mechanisms, such as the disease transitioning to other cytokine pathways, may contribute to the observed variable nonresponse [352]. In anti-TNF therapy-resistant IBD, alternative drivers of chronic inflammation, particularly IL-23 and Oncostatin M (OSM), have been identified. OSM, a member of the IL-6 family that primarily signals into mesenchymal cells, is considered a key player in this context. The shift from TNF to OSM as a dominant driver of chronic inflammation in individuals with established disease and anti-TNF therapy resistance suggests distinct downstream pathways elicited by these cytokines in mesenchymal cells. This highlights OSM as an interesting cellular target in IBD [24,353]. 

Neurath et al. proposed that cytokine networks in the inflamed mucosa of CD patients undergo changes influenced by varying cytokine production patterns throughout the course of the disease. Additionally, recent trials have demonstrated the heterogeneity in the pathogenesis of IBD, indicating that TNF-independent cytokine signaling pathways, independent of the TNF signaling pathways, also contribute to the development of resistance to anti-TNF therapies [339]. Patients with CD who respond to anti-TNF treatment exhibit higher expression of TNFR2 on mucosal T cells than nonresponders before therapy initiation. Molecular mechanisms driving IL-23-mediated resistance against anti-TNF therapy involve the upregulation of mucosal IL23p19, IL23R, and IL17A, but not IL-12p40, in CD patients resistant to anti-TNF therapy during ongoing treatment. The expansion of apoptosis-resistant intestinal TNFR2+IL23R+ T cells is associated with anti-TNF resistance in CD. These cells express gut-tropic integrins α4β7, contributing to the perpetuation of mucosal intestinal inflammation. The identification of dual IFN-γ- and IL-17-producing T cells highlights the dynamic nature of the cytokine network in CD, developing through an IL-23-driven compensatory inflammatory pathway upon TNF blockade. CD14+ macrophages were identified as potent producers of IL-23 in CD, and these effects were absent in responders to anti-TNF therapy [339]. Targeting IL-23 may be justified in CD patients resistant to anti-TNF therapy due to the identification of apoptosis-resistant intestinal TNFR2+IL23R+ T cells and the dynamic cytokine network, indicating a potential TNF-independent pathway involving IL-23 [339].

Recently, Guo et al. found that high pretreatment OSM concentrations identify IBD patients at-risk of anti-TNF nonresponse at 1 year, as well as other deleterious clinical outcomes, providing further support that OSM drives intestinal inflammation and may predict response to anti-TNF therapy in patients with IBD [354]. Elevated intestinal expression of OSM and OSM receptor (OSMR) was strongly associated with a lack of response to TNF-neutralizing therapy [353].

Aguilar et al. analyzed the CELEST study, a randomized controlled trial substudy investigating cell-specific mechanisms of the JAK1 inhibitor upadacitinib in the intestinal mucosa of CD patients. The study concluded that upadacitinib modulates inflammatory pathways in mucosal lesions of anti-TNF-refractory CD patients, affecting inflammatory fibroblast and interferon-γ-expressing cytotoxic T cell compartments. This study represents the first description of the molecular response to JAK1 inhibition in IBD, highlighting differential effects compared to anti-TNF treatment [355]. Additional considerations involve the clinical efficacy of upadacitinib and risankizumab in addressing reduced IBD anti-TNF-α inadequate response mechanisms. Moreover, tofacitinib, an inhibitor targeting JAKs upstream of STAT pathways, has received approval for use in UC [356]. Recent data indicate that treatment with selective IL-23 inhibitors, such as risankizumab, results in significantly high response rates among CD patients who were refractory to previous anti-TNF therapy, underscoring the significance of IL-23 in anti-TNF-refractory intestinal inflammation [330]. Furthermore, Wang and colleagues concluded that integrative transcriptomics delineates cellular and molecular mechanisms associated with clinical efficacy of upadacitinib and risankizmab in IBD subjects who are inadequate responders to TNF inhibitors that hold the potential to optimize clinical responses to therapy [338]. In the GEMINI trials, vedolizumab showed comparable efficacy in patients previously exposed to anti-TNF agents, regardless of whether they experienced primary or secondary nonresponse [357]. A recent systematic review and meta-analysis indicated that the treatment response to ustekinumab was less favorable in individuals who were primary nonresponders to anti-TNF therapy compared to those who experienced secondary nonresponse (relative risk [RR] 0.64 [0.52–0.80]) [358]. In a recent multicenter retrospective study conducted by Kassouri et al., the outcomes of patients with late-stage Crohn’s disease (CD) who had previously failed treatment with one anti-TNF agent and subsequently received either vedolizumab or ustekinumab as a second-line therapy were reported. After 48 weeks on a third line of biologic therapy, the remission rate was 30.7%, while the surgery rate was 23.5% [344]. Interestingly, emerging data suggest that an IL-23 blockade with agents such as brazikumab, mirikizumab, guselkumab, and risankizumab may be the preferred approach for optimized management in patients previously exposed to TNF antagonists, regardless of primary or secondary loss of response [337].

## 7. Emerging Strategies Targeted at Cytokine Networks and Biomarkers

Sphingosine-1-phosphate (S1P) and S1P receptors (S1PR) have been well characterized in immune trafficking as part of cytokine-related signaling proteins. Modulation on S1PR is an interesting target for the treatment of IBD. Ozanimod and etrasimod, a new S1PR modulator, is a promising new oral treatment option for IBD. In moderately to severely active ulcerative colitis (UC), phase 2b and/or phase 3 studies achieved primary endpoints for S1P receptor agonists such as estrarimod and ozanimod [359,360,361]. Currently, there are ongoing trials investigating other new-generation S1PR1 modulators.

The IL-36 family belongs to a larger IL-1 superfamily and consists of three agonists (IL-36α/β/γ), one antagonist (IL-36Ra), one cognate (IL-36R), and one accessory protein (IL-1RAcP). Antagonist IL-36Ra inhibits the signaling by binding to IL-36R and preventing recruitment of IL-1RAcP [362]. Spesolimab as an IL-36 receptor antagonist was developed by Boehringer Ingelheim for the treatment of various immune-mediated disorders. In 2022, spesolimab was approved in the USA for the treatment of generalized pustular psoriasis in adults [363]. Clinical trials evaluated effects of therapeutic targeting IL-36R on the outcomes of UC. Early clinical data from a randomized, double-blind, multicenter, phase II trial that is currently under way (NCT03482635) indicate low rates of clinical remission in patients with moderate-to-severe UC [364]. Another randomized, double-blind, phase IIa trial (NCT03123120) found that spesolimab did not induce mucosal healing in patients with mild-to-moderate UC receiving stable TNF-α antagonists [364]. In an open-label, single-arm, phase IIa exploratory trial (NCT03100864), spesolimab had a limited effect on gene expression in patients with moderate-to-severe, active UC [364]. More research is required to determine the therapeutic potential of IL-36R signaling modulation in CD patients. Spesolimab is being developed for the treatment of CD (at phase II) NCT03752970; EudraCT2017-003090-34, NCT04362254; EudraCT2019-001673-93, NCT05013385; EudraCT2020-005770-99 [363]. Endogenous agonists function as proinflammatory cytokines, and IL-36 signaling directly stimulates mesenchymal cells, triggering a profibrotic transcriptional program and promoting the secretion of profibrotic mediators [202]. Hence, targeting IL-36R blockade has been suggested as a potential therapeutic strategy for treating profibrotic disorders [365].

Quisovalimab (AVTX-002 (AEVI-002, KHK-252067, SAR-252067)) is under clinical development in Phase II for the treatment of CD. It is administered through a subcutaneous route. It is a fully human IgG4 anti-LIGHT (TNGSF14) monoclonal antibody. VTX002 neutralizes both soluble and membrane LIGHT. The safety profile and pharmacokinetic properties of the fully human monoclonal anti-LIGHT antibody, SAR252067, were evaluated in healthy volunteers in phase 1a studies as a potential treatment for diseases related to LIGHT-mediated mucosal inflammation [366]. LIGHT, a member of the TNF superfamily, is potentially involved in mucosal inflammation associated with IBD. Quisovalimab binds to LIGHT and prevents it from interacting with its receptors on T cells [367].

The pathogenesis of IBD-associated fibrosis involves various factors, including mesenchymal cells, cytokines (especially TGF-β), growth factors, microRNAs, intestinal microbiome, matrix stiffness, and mesenteric adipocytes. Despite extensive preclinical studies, there is currently no available antifibrotic therapy to prevent or reverse intestinal fibrosis in CD [368]. Targeting TGF-β signaling pathways is considered the most promising approach for antifibrotic therapy, as TGF-β is the principal molecular mediator of fibrogenesis in IBD-associated fibrosis [368].

TNF, through its two receptors, TNFR1 and TNFR2, exhibits dual effects on cell fate, promoting either cell death or survival in different cell types. The proinflammatory and anti-inflammatory properties of TNF are largely determined by its binding to TNFR1 and TNFR2, respectively [369]. Moreover, the two receptors differ in the intracellular signaling pathways they activate, resulting in distinct cellular responses [370]. In general, membrane-bound TNF stimulates cell survival and proliferation through TNFR2 activation, whereas soluble TNF initiates apoptotic and proinflammatory signals via TNFR1 [371]. The anti-inflammatory effects of TNF are explained by its capacity to increase the proliferation, stability, and suppressive function of FOXP3+ Treg cells via TNFR2 signaling [372]. The immunoregulatory role of TNF and the ability to neutralize it regardless of its downstream functions may contribute to the failure of anti-TNF therapies. This limitation in clinical efficacy could be attributed to the differences between TNFR1 and TNFR2 signaling and the diverse effects of TNF on various immune cells, including FOXP3+ Treg cells. Several in vivo studies suggest the potential of selectively targeting TNFRs to restore homeostasis in different animal models of inflammatory diseases. Pegoretti et al. highlight the complexity of TNF signaling and propose that a timely balance of selective activation and inhibition of TNFRs is necessary for therapeutic effects. Sequential treatment with a TNFR2 agonist and a TNFR1 antagonist has shown improved outcomes in a humanized mouse model [370]. Research suggests that TNFα-TNFR1 and TNFα-TNFR2 play differential roles in the differentiation and function of CD4+Foxp3+ induced Treg cells in autoimmune diseases. Exogenous TNFα may enhance the differentiation and function of induced Treg cells via TNFR2 signaling. In certain autoimmune diseases, there may be downregulation of TNFR2 expression on Treg cells, accompanied by an increased level of TNFR1. Consequently, TNFR2 agonists or TNFR1-specific antagonists hold potential promise for clinical applications in treating patients with IBD [373]. It is suggested that the next generation of anti-TNF targeted drugs may involve antagonists of TNFR1 that selectively block the binding of TNF to TNFR1 and agonists of TNFR2. This approach could potentially be more effective and have fewer adverse effects compared to classical anti-TNF drugs [374]. In the early 1990s, it has been described that LTα and LTβ form LTα2β and LTαβ2 heterotrimers, which bind to TNFR1 and LTβR, respectively. Afterwards, the LTαβ2-LTβR system was intensively studied, while the LTα2β-TNFR1 interaction has been ignored to date, presumably due to the fact that at the time of identification of the LTα2β-TNFR1 interaction, one knew already two ligands for TNFR1, namely TNF and LTα. LTα2β interacts not only with TNFR1 but also with TNFR2 and membrane-bound LTα2β (memLTα2β), and despite its asymmetric structure, stimulates TNFR1 and TNFR2 signaling. Not surprising in view of its ability to interact with TNFR2, LTα2β is inhibited by Etanercept, which is approved for the treatment of rheumatoid arthritis and also inhibits TNF and LTα [375].

IL-6 or its receptor is being considered as a candidate for targeted biological therapy in IBD. University researchers are investigating IL-6 signaling mechanisms and its biological effects, while the company is focused on developing and characterizing IL-6 inhibitors for potential use in treating autoimmune diseases [376,377]. IL-6 signals through a complex of IL-6 R alpha and gp130. gp130 is also a component of the receptors for CLC, CNTF, CT-1, IL-11, IL-27, LIF, and oncostatin M. Soluble forms of IL-6 R alpha are generated by both alternative splicing and proteolytic cleavage. In a mechanism known as trans-signaling, complexes of soluble IL-6 and IL-6 R alpha elicit responses from gp130-expressing cells that lack cell surface IL-6 R alpha. Deletion of IL-6 exacerbates colitis and induces systemic inflammation in IL-10-deficient mice. Complete IL-6 blockade significantly worsens gut inflammation in IL-10-/- mice, partly by suppressing Treg/CTLA-4 and promoting the IL-1β/Th2 pathway. The double mutant exhibits signs of severe systemic inflammation. These findings highlight a new role for IL-6, suggesting caution in targeting IL-6 in IBD patients, especially those with IL-10 signaling defects [216]. Furthermore, IL-4-conditioned MDMs were more effective at supporting Th2 differentiation and inhibiting Th1 and Th17 differentiation of CD4+ T cells. Together, these studies demonstrated that fecal bacteria from CD patients presented enhanced capacity to upregulate pattern-recognition molecules in macrophages, which could be repressed by IL-4 [378]. Terabe et al. suggest that suppression of the proliferation of pathogenic CD4+ T cells is the major mode of action of biological agents for colitis therapy. Anti-IL-6R mAb might have benefits in CD patients with Th17 dominance and impaired Treg frequency [379]. The therapeutic potential of IL-6 signaling blockade for CD, anti-IL-6R monoclonal antibody (mAb), was introduced to various murine models of colitis. These results strongly suggest that specific targeting of the IL-6/sIL-6R pathway will be a promising new approach for the treatment of CD [380]. Characterization of HZ0412a, a novel potent humanized anti-6 receptor antibody that blocks IL-6R binding togp130 demonstrated a favorable anti-IL-6R antibody profile, including effectively binding to IL-6R and successfully antagonizing the interaction of IL-6R and gp130. HZ0412a binds with high affinity to the human IL-6 receptor, with a binding affinity (KD) of 14.5 nM. This binding profile is almost three-fold stronger than tocilizumab, which has been approved in anti-IL-6 therapies [381].

Canakinumab is a human anti-IL-1β monoclonal antibody [382]. This study shows the use of canakinumab in autoinflammatory VEO-IBD. Canakinumab is well tolerated and can be a good therapeutic option in this subset of patients. Clinical response was achieved in 17/19 (89%) patients by six months [383]. England et al. discovered tozorakimab, a novel high-affinity IL-33 antibody that inhibits signaling through both the IL-33red–ST2 and IL-33ox–RAGE/EGFR pathways. Unlike anti-ST2 therapeutics targeting only IL-33red–ST2 signaling, tozorakimab’s mechanism of action is distinct [384]. There are three anti-IL-17 monoclonal antibodies used in IBD, secukinumab, ixekizumab, and brodalumab. They are blocking the binding of IL-17A to IL-17 R, inhibiting the proinflammatory effects of IL-17A downstream [385]; however, the efficacy is contradictory.

Oncostatin M (OSM) and its receptor (OSMR) are elevated in the inflamed intestines of IBD patients, correlating strongly with disease severity [353]. OSM may amplify inflammation and drive chronicity by attracting mononuclear phagocytes (MNPs) and T cells. The influence of OSM on fibrotic processes via the stromal compartment remains unclear. Notably, OSM can bind to extracellular matrix (ECM) components, protecting it from degradation and maintaining biological activity over extended periods [353]. Cao et al. showed that serum levels of OSM were positively related to disease activity. It was notably elevated in patients with active CD and moderate-to-severe UC compared to those in remission. Clinical nonresponders to anti-TNF had higher serum OSM expression than responders to anti-TNF [386]. Verstockt predicted that elevated colonic OSM and OSMR were associated with a worse disease prognosis, for example, the requirement of biologic therapy within 2 years after diagnosis [387]. And as written above, plasma OSM concentrations may represent an important biomarker of a lack of response to anti-TNF therapy [354]. The OSM pathway may synergize with those activated by TNF, suggesting that a dual blockade of both TNF and OSM could be beneficial for some individuals.

Fibrotic complications, including strictures, are significant manifestations of IBD. These complications narrow certain parts of the intestine, leading to structural and functional damage, significantly impacting patients’ quality of life [5]. Myofibroblasts are key effector cells in the formation of intestinal fibrosis [388]. IL-34 is a novel cytokine that was identified in 2008 in a comprehensive proteomic analysis as a tissue-specific ligand of CSF-1 receptor [389]. IL-34 demonstrated profibrotic properties in vitro by inducing the production of collagens COL1A1 and COL3A1 in fibroblasts through a p38 MAP kinase-dependent mechanism and enhancing wound healing [195]. IL-36 family belongs to a larger IL-1 superfamily [362]. Signaling via IL36 receptor is essential to induce and maintain tissue fibrosis in murine models, because anti-IL36R antibodies reverse established tissue fibrosis in chronic intestinal inflammation [203]. TGF-β1 is a secreted profibrotic cytokine, which intricately controls a plethora of physiological and pathological processes [390]. In IBD, IL-34 promotes fibrosis by regulating various cells through the activation of canonical Smad signaling pathways. This involvement contributes to the development of epithelial-to-mesenchymal transition in epithelial cells, fibroblast proliferation, and the transformation of fibroblasts and smooth muscle cells into myofibroblasts [5]. Wang et al. found out DSS-induced IBD-related intestinal fibrosis in mice by inhibiting TGF-β-induced intestinal fibroblast proliferation, migration, and activation [391]. Another study by Xu et al. shows that TL1A affects epithelial to mesenchymal transition (EMT) in IBD patients via the TGF-β/Smad3 pathway, causing colonic fibrosis and inflammatory responses. TL1A is capable of increasing the barrier permeability of TNF-α-induced Caco-2 cell and reducing function of tight junction protein (TJ) through the myosin light chain kinase/p-myosin II regulatory light chain (MLCK/p-MLC) pathway and the LPS-mediated myeloid differentiation factor 88/TNF receptor-associated factor-6 (MyD88/TRAF6) pathway, which further damage the intestinal mucosal barrier [392]. Jun et al. studied fibrosis in TLR 4 TLR4 gene-deficient mice indicating that those mice exhibit a reduced colonic inflammation as well as a decrease in the infiltration, thereby resulting in reduced collagen deposition and intestinal fibrosis [393]. Interestingly metformin is protective against intestinal fibrosis induced by TNBS or DSS in vivo and inhibits activation and collagen synthesis in colon fibroblasts [394]. A study by Buterra demonstrated the involvement of CD147 in the intestinal fibrosis process and the ability of AC-73 (a small molecule able to inhibit CD147 signaling and induce autophagy) administration to inhibit the fibrogenic process in TNBS chronic colitis. They observed a progressive increase in CD147 protein expression during the development of intestinal fibrosis, suggesting that CD147 may have an important role in fibrogenesis. Early use of the CD147 signaling inhibitor reduces fibrosis by modulating multiple processes involving inhibition of the ERK1/2 and STAT3 pathways, as well as autophagy induction. CD147 represents a new potential target to reduce fibrosis in IBD, and its inhibitor, AC-73, might represent a possible antifibrotic drug in the management of fibrostenosing CD [395]. Xie et al. have demonstrated that integrin αvβ6 expression was significantly increased in the stenotic region compared to the normal region. Their data show the role of integrin αvβ6 in intestinal fibrosis, identifying αvβ6 as a potential promising novel therapeutic target to prevent intestinal fibrosis. Integrin αvβ6-associated intestinal fibrosis is related to the activation of the FAK/AKT pathway [396]. Weder et al. found out that in the human terminal ileum, increased expression of fibrosis markers was accompanied by an increase in GPR4 expression. A positive correlation between the expression of procollagens and GPR4 was observed. GPR4 deficiency was associated with a decrease in angiogenesis and fibrogenesis evidenced by decreased vessel length and expression of Edn, Vegfα, and procollagens [397]. A study by Lee et al. demonstrated that the combination of pentoxifylline (PTX) and vitamin E had notable antifibrotic effects in human primary intestinal myofibroblasts and in in vivo inflammatory bowel disease (IBD) models. Long-term therapy with PTX and vitamin E together was shown to decelerate the advancement of intestinal fibrosis, preserving gut resilience and potentially improving the quality of life for individuals with IBD. The results also suggest that vitamin E supplementation may contribute to maintaining intestinal flexibility in IBD patients [398].

A study by Liso et al. suggests that novel targeted drugs, particularly anti-IL-1 strategies like anakinra, may offer a more effective therapeutic option for primary nonresponders to anti-TNF therapy in ulcerative colitis (UC). The researchers analyzed gut mucosal biopsy specimens and circulating cytokine profiles of 30 UC patients, finding that approximately 75% of primary nonresponders had abundant IL1β in both serum and local tissues. In Winnie-TNF-KO mice, the administration of anakinra efficiently reduced the histologic score of the distal colon, a common site of inflammation in these mice. This research provides new insights and alternative approaches for UC patients who do not respond well to anti-TNF therapy [399]. Di Martino et al. conducted an interesting study in an animal model, focusing on TNF-like weak inducer of apoptosis (TWEAK) and its receptor, fibroblast growth factor–inducible 14 (Fn14). They found out that TWEAK and Fn14 are upregulated in CD, and also mediate experimental CD-like ileitis, by regulation of multiple innate and adaptive cellular pathways. Therefore, TWEAK/Fn14 may represent a novel therapeutic target for the treatment of CD [400]. Furthermore, researchers indicated that treating zebrafish or cultured cells with a gp130-blocking drug inhibits activation of inflammatory cells. This study suggests that drugs targeting IL-13, when used in conjunction with anti-TNF therapy, might be effective treatments for people with CD resulting from NOD2 risk variants [401]. Therapy with ABX464 50 mg once daily appeared to be safe and well tolerated in patients with UC. After 8 weeks of treatment, ABX464 appeared to be more effective than placebo in achieving endoscopic improvement and a reduction in the MCS (Mayo Clinic score) and pMCS (Partial Mayo Score). Maintenance therapy with ABX464 was effective in sustaining remission, and even inducing remission in additional patients [402]. A study by Schreiber et al. suggests that blockade of IL6 trans-signaling holds great promise for the therapy of IBD and should undergo full clinical development as a new therapy for IBD [403]. Yang et al. depicted the role of GPR120 in suppressing intestinal CD4+ T-cell induction of colitis through promoting the production of IL-10, suggesting the GPR120 agonist as a potential therapeutic target for treating IBD [233]; thus, targeting IL-10 may offer an alternative approach. Promoting anti-inflammatory cytokine responses using IL-2, IL-10, or TGF has shown promise in preclinical and clinical studies, but lacks rigorous evaluation in clinical trials [24]. Limited biomarkers exist for predicting IBD before diagnosis. In UC patients, anti-integrin αvβ6 autoantibodies precede clinical diagnosis by up to 10 years and are associated with adverse UC-related outcomes [404].

## 8. Conclusions and Future Direction

One of our challenges has been the unraveling of cytokines and signaling pathways in IBD; however, the exact etiology of intestinal inflammation in CD and UC is not fully elucidated, though it is postulated that modulation of multiple cytokine networks and dysregulation of any components of this network must be regarded as a critical mechanism driving the initiation, resolution and perpetuation of inflammation. Cytokines and cytokine receptors are considered to be excellent targets for medicinal biotherapeutics. Using anticytokine drugs are extremely effective in inducing mucosal healing in IBD, however, individual variation in drug response has been identified. Novel anticytokine agents or combination therapies might prove useful in this respect. Unraveling cytokine networks, the cell–cell interactions, the direct pathogenetic relevance, and immune consequences of dysregulation in the gut appears to offer us a new direction for diagnostic and therapeutic approaches and improved clinical outcomes in IBD.

## Figures and Tables

**Figure 1 biomedicines-11-03229-f001:**
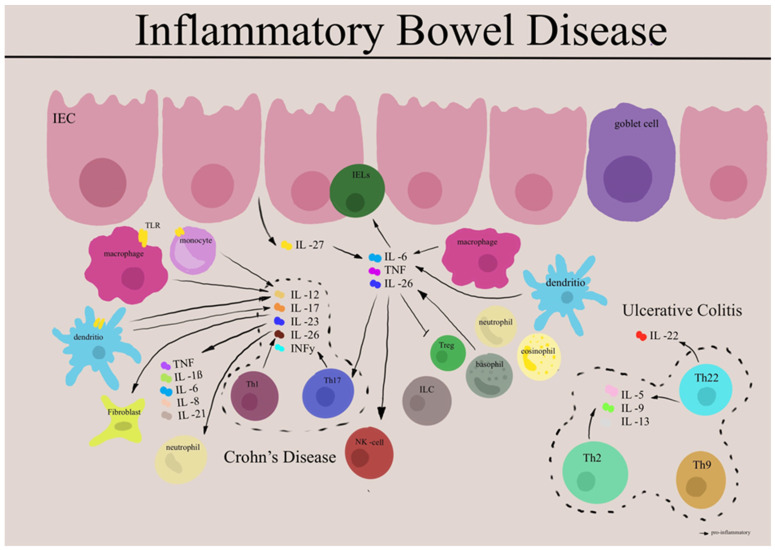
Complex network of cytokines in pathophysiology of IBD. IEC: intestinal epithelial cells, IELs: intraepithelial lymphocytes, NK cell: natural killer cells, TLR: toll-like receptor, TNF: tumor necrosis factor, ILC: innate lymphoid cells, IL: interleukins, Th: T-helper cells, INFγ: interferon gamma. Adapted from [40].

**Table 1 biomedicines-11-03229-t001:** Cytokine family—groups, subgroups, and effect in IBD.

Cytokine Family	Subgroups	Effect in IBD
Tumor necrosis factor	TNF-α TNF-β	proinflammatory activate nonspecific immunity adhesive molecular expression on endothelial surfaces may lead to apoptosis similar effect, produced by T and B cells
Interferons	INF α, β, ω INF γ	antiviral immunity antiproliferative effect antitumor activity response to anticellular pathogens
Colony stimulating factors	G-CSF, GM-CSF, M-CSF	stimulate proliferation and maturation of myeloid cells
TGFβ		stimulate growth of fibroblasts and extracellular matrix production
Interleukins	IL-1, IL-18, IL-33, IL-36, IL-38	proinflammatory
	IL-4, IL-6, IL-10, IL-11, IL-13, IL-22	Anti-inflammatory
	IL-9	pleiotropic

TNF-α: tumor necrosis factor alpha, TNF-β: tumor necrosis factor beta, INF: interferon, G-CSF: granulocyte colony stimulating factor, GM—CSF: granulocyte-macrophage colony stimulating factor, M-CSF: macrophage colony stimulating factor, TGF-β: transforming growth factor beta, IL: interleukin. Adapted from [34].

**Table 2 biomedicines-11-03229-t002:** Major interleukins and function in IBD.

Interleukin	Th Subset	Function in IBD	Therapeutics
IL-1	Th	Mediator of inflammation and tissue damage	Canakinumab (anti IL-1β)
IL-17	Th17	T-cell activation to neutrophil mobilization and activation	Secukinumab, Ixekizumab, Brodalizumab
IL-18	Th17, Tregs	Intraepithelial cells proliferation, tissue regeneration, production of proinflammatory cytokines	-
IL-33	Tregs	Type 2 immune response, intraepithelial cells differentiation, intestinal inflammation	Tozarakimab
IL-4	Th2	Induces IgE class-switch recombination in B cells; induces Th2 differentiation and IL-13 expression	Pasolizumab (anti IL-4), Altrakincept (soluble Il-4R)
IL-6	Th17	IEC proliferation and repair, crypt homeostasis	Olamkicept
IL-10	Tregs	Inhibits proinflammatory cytokine expression by innate/adaptive immune cells; STAT3-dependent signaling	Recombinant IL-10
IL-11	Macrophages	Activation of JAK/STAT signaling, tumor cell survival	-
IL-13	Th 2	Impairs epithelial barrier function; promotes mucosal fibrosis via induction of TGFβ1 expression	Anrukizumab, Lebrikizumab, Tralokizumab
IL-22	Th17, Th 22	Enforces epithelial barrier function; stimulates expression of antimicrobial peptides, defensins	Fezankizumab
IL-9	Th9	Impairs mucosal wound healing; regulates epithelial cell proliferation, barrier function	Enokizumab

Adapted from [129].

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
