# Peer review of "A Narrative Review of Cytokine Networks: Pathophysiological and Therapeutic Implications for Inflammatory Bowel Disease Pathogenesis"

_biomedicines, 2023, doi:10.3390/biomedicines11123229_

Round 1
Reviewer 1 Report
Comments and Suggestions for Authors
This paper presented an integrative review of cytokine networks in inflammatory bowel disease (IBD). The authors provided a detailed description of the functions and complex effects of cytokines involved in IBD, including an explanation of the gut immune system. In addition, the functions of individual cytokines were described in detail while providing enough evidences. They also showed the targeting therapies in IBD. Anti-integrins, as anti-lymphocyte trafficking agents, IL-23 or IL-12/23 inhibitors, and anti-TNF-α agents were introduced as the targeting therapies, showing their mechanisms and clinical trials. This review report is well written. It will be of interest to people in the field. Each section was described in considerable detail, so the figure and tables help the reader understand. Therefore, it would be better to add a table or figure of targeting therapy agents in the section 6.
Author Response
Hello, thank you for your opinion. As written bellow, we will consider to make table of targeting therapy agents mentioned in section 6. Now we did not do that, because it is well known agents, and many figures were made previously.
Reviewer 2 Report
Comments and Suggestions for Authors
- - Use oxford comma in the whole text
- -“flora”
Use microbiome or microbiota terms
- -“For these reasons, these properties suggest that IL-12 family cytokines have a key role in the regulation of intestinal homeostasis, and ultimately, the pathogenesis of IBD, and they have become potential targets for inhibiting the pathogenesis of inflammatory bowel disorders”
Why blocking anti-IL/23 alone in spite of blocking anti-IL12 and anti-IL23 is more effective?
- -“IL-17 plays a critical role in inflammatory and immune mechanisms through which IL-17 is considered a molecular target for the development of novel IL-17A blocking agents for the treament of IBD”
Why anti-IL17 drugs increase the risk of IBD (see “Vernero M et al. New Onset of Inflammatory Bowel Disease in Three Patients Undergoing IL-17A Inhibitor Secukinumab: A Case Series. Am J Gastroenterol. 2019 Jan;114(1):179-180. doi: 10.1038/s41395-018-0422-z. PMID: 30429591.)
- -Add dupilumab in Table 2 and discuss its potential role in IBD
- -“IL-23 (brazikumab, risankizumab, and mirikizumab)”
Cite guselkumab, tildrakizumab
- -Explain why anti-IL/6 drugs seem not work in IBD
Comments on the Quality of English LanguageGood
Author Response
Hello, thank you for your review and interesting opinions.
We tried to change everything you suggested down below. Your recommendations were cited in chapters.
Reviewer 3 Report
Comments and Suggestions for Authors
Review for the manuscript “A narrative review of cytokine networks: pathophysiological and therapeutic implications for inflammatory bowel disease pathogenesis.”
Dear Editor, thank you for the invitation to review this very interesting manuscript. After careful evaluation, I have some comments and suggestions before the publication process can be continued.
Overall comments: This is a study where authors “summarize the main roles of substantial cytokines in IBD related to homeostatic tissue functions and the remodeling of cytokine networks in IBD, which may be specifically valuable for successful cytokine-targeted the apies via marketed products.”
ABSTRACT
In lines 19- 25 we can read “…Cytokines and their receptors are validated targets for multiple therapeutic areas, we review the current strategies for therapeutic intervention and developing cytokine-targeted therapies. New biologics have shown efficacy in the last decades for the management of IBD, unfortunately, many patients are nonresponsive or develop therapy-resistance over time creating a need for novel therapeutics. Thus, the treatment options for IBD beyond the immune-modifying anti -TNF agents or combination therapies are expanding rapidly. Further studies are needed to fully understand the immune response, networks of cytokines, and the direct pathogenetic relevance regarding individually tailored, safe and efficient targeted-biotherapeutics”. I suggest changing for “…Cytokines and their receptors are validated targets for multiple therapeutic areas. In this study, we review the current strategies for therapeutic intervention and developing cytokine-targeted therapies. New biologics have shown efficacy in the last decades for the management of IBD, unfortunately, many patients are nonresponsive or develop therapy-resistance over time creating a need for novel therapeutics. Thus, the treatment options for IBD beyond the immune-modifying anti -TNF agents or combination therapies are expanding rapidly. Further studies are needed to fully understand the immune response, networks of cytokines, and the direct pathogenetic relevance regarding individually tailored, safe and efficient targeted-biotherapeutics”
INTRODUCTION
In lines 31-36 we can read: “ Inflammatory bowel diseases (IBD), encompasses Crohn’s disease (CD) and ulcera-30 tive colitis (UC). IBD is a chronic relapsing immune-mediated disease that is likely to oc-31 cur in early childhood to beyond the sixth decade of life and is unfortunately incurable. Previous systematic reviews described rising incidence and prevalence of IBD among both children and adults around the world and data are emerging from regions where it was previously thought to be uncommon 1 2 The origin of this disease is not entirely 35 clear and several involved mechanisms have been postulated such as genetics, defects in 36 a number of cellular pathways including the dysregulation of homeostasis, loss of epi thelial barrier integrity and tolerance to the gut microbioma, and environmental exposures among other processes3 4 5 6 7 8 The immunological...” Please check citation of the references. A period is necessary at the end of each sentence. Please correct and check other sentences along with all the text.
Furthermore, when using several references in sequence, as in line 39, do it as follows: instead of “3,4,5,6,7,8” use “3-8”.
Check the style of the references in the text. MDPI has particular style guidelines.
DISCUSSION
In line 97, we can see “components of the immune system. Table 1. Our knowledge of immune-mediated inflammation has been…” Please explain one sentence with “Table 1.”
Please improve the description of the title of Table 1 and define all the abbreviations included in this table.
In line 517, for example, the authors use Interleukins (IL). However, this abbreviation was used before in the text. For this reason, it is not necessary to do it again. When an acronym is defined, please use only it in the rest of the text. Check this for all the abbreviations included in the text. However, it is always necessary to define abbreviations in tables, even if they have been described in the text.
Figure 1 is not mentioned in the text. I suggest that the authors build a new and better-quality figure.
The title of the figures should come at the bottom of them. Moreover, in Figure 1, there are abbreviations missing in the legend, and the authors say that it was “Adapted from: Mahapatro, Mousumi et al. “Cytokine-Mediated Crosstalk between Immune Cells and Epithelial Cells in the Gut.” This citation is not complete. Please include the year of the publication or use number for this citation.
In line 1233 we see: “…parent role in several inflammation-related processes makes them appealing targets for 1233 the development of IBD therapies 4 7…” Are the citations 4 and 7 or 47?
I appreciate the item about the limitations of the study.
CONCLUSION
This section is adequately described. However, I suggest including the limitations of this review.
REFERENCES
As pointed out above, check the style of the references in the text and the References section. MDPI has particular style guidelines.
FINAL COMMENTS:
I suggest that the authors double-check punctuation and grammar. I believe that professional editing services should edit it.
Comments on the Quality of English LanguageMajor corrections are necessary.
Author Response
Hello, thank you for your rewiev of our article. I tried to implement and change all what you mentioned bellow.
For the language, we were in contact with specialist from journal. English proof reading (changes of grammar and punctuations) will be done in future.
Round 2
Reviewer 3 Report
Comments and Suggestions for Authors
Dear authors,
Thank you for the revised version.
I wish good luck with this ms.
Minor.